**1. Title page**
**The effects of burning and grazing on soil carbon dynamics in managed Peruvian tropical**
**montane grasslands**
Viktoria Oliver[1,2]*, Imma Oliveras[3], Jose Kala[4], Rebecca Lever[5,2], Yit Arn Teh [1,2]
[1] School of Biological Sciences, University of Aberdeen, Cruickshank Building, St. Machar Drive,
AB24 3UU Aberdeen, UK.
[2] Formerly at the School of Geography and Geosciences, University of St Andrews, UK
[3] Environmental Change Institute, School of Geography and the Environment, University of
Oxford. South Parks Road, OX13QY Oxford, UK.
[4] Universidad de Santo Antonio Abad del Cusco, Cusco, Peru.
[5] Department of Life & Environmental Sciences, University of California, Merced 5200 North
Lake Rd. Merced, CA 95343, United States.
* Corresponding author: v.oliver@abdn.ac.uk
*Running title:* Tropical montane grassland soil carbon dynamics
*Keywords:* Andean montane grasslands, soil respiration, fire, grazing, puna, soil carbon, land-
use activities, soil density fractionation.

## 2. Abstract

Montane tropical soils are a large carbon (C) reservoir, acting as both a source and a sink of $CO_2$. Enhanced $CO_2$ emissions originate, in large part, from the decomposition and losses of soil organic matter (SOM) following anthropogenic disturbances. Therefore, quantitative knowledge of the stabilization and decomposition of SOM is necessary in order to understand, assess and predict the impact of land management in the tropics. In particular, labile SOM is an early and sensitive indicator of how SOM responds to changes in land use and management practices, which could have major implications for long term carbon storage and rising atmospheric $CO_2$ concentrations. The aim of this study was to investigate the impacts of grazing and fire history on soil C dynamics in the Peruvian montane grasslands; an understudied ecosystem, which covers approximately a quarter of the land area in Peru. A density fractionation method was used to quantify the labile and stable organic matter pools, along with soil $CO_2$ flux and decomposition measurements. Grazing and burning together significantly increased soil $CO_2$ fluxes and decomposition rates and reduced temperature as a driver. Although there was no significant effect of land use on total soil C stocks, the combination of burning and grazing decreased the proportion of C in the free LF, especially at the lower depths (10-20 and 20-30 cm). In the control soils, 20 % of the material recovered was in the free LF, which contained 30 % of the soil C content. In comparison, the burnt-grazed soil, had the smallest recovery of the free LF (10 %) and a significantly lower C content (14 %). The burnt soils had a much higher proportion of C in the occluded LF (12%) compared to the not-burnt soils (7%) and there was no significant difference among the treatments in the heavy F (~ 70%). The synergistic effect of burning and grazing caused changes to the soil C dynamics. $CO_2$ fluxes were increased and the dominant temperature driver was obscured by some other process, such as changes in plant C and N allocation. In addition, the free LF was reduced when these two anthropogenic activities took place on the same site. Most likely a result of reduced detritus being incorporated into the soil. A positive finding from this study is that the total soil C stocks were not significantly affected and the long term (+10 years) C storage in the occluded LF and heavy F were not negatively impacted. Possibly this is because of low intensity fire, fire-resilient grasses and the grazing pressure is below the threshold to cause severe degradation.

**Commented [OV2]:** 'Negative effects' removed

## 3. Introduction

High altitudinal montane grasslands (3200 - 4500 m a.s.l) account for a major proportion of land cover in the Andes, particularly in Peru, where they make-up approximately 25 % of land cover (Feeley and Silman 2010). Every year, especially in the dry season, large areas of these grasslands are burned to support traditional cattle grazing, which has been apparent since the early 1500s (Luteyn 1992). Fires for agricultural clearing and maintenance of these highly productive forage grasses is of considerable importance in these ecosystems and for the livelihood of the local people (Sarmiento and Frolich 2002). To some extent, this natural system is tolerant of these management practices (Ramsay 1992). However, in recent years, it has become apparent that the combination of global warming and the considerable pressure from agricultural expansion have resulted in increased fire occurrence and subsequent destruction of tropical montane cloud forest (Cochrane and Ryan 2009). Evidence of fire scars and charcoal deposits along the forest-puna tree line demonstrate a gradual encroachment into the adjacent tropical montane cloud forest (Di Pasquale *et al.* 2008).

Previous research in these Andean montane grasslands have measured large soil C stores, (Gibbon *et al.* 2010; Oliveras *et al.* 2014b). However, despite the concern on the effects of land management practices, there are very few studies on soil C dynamics in this tropical region of the Peruvian Andes. It is particularly unclear how land management affects the soil C dynamics and sequestration potential under the influence of grazing and burning. For example, (Oliveras *et al.* 2014b) , found that grazing and fire in montane grasslands resulted in decreased net primary productivity, but there were no differences between these two disturbances. Studies in other montane grasslands have found that an increase in the frequency of fire events can reduce the amount of soil organic matter (SOM) in the top soil (Knicker 2007), or it may increase the biomass growth period afterwards, causing more detritus to accumulate in the upper soil layers (Ojima *et al.* 1994).

SOM influences many soil functions and occupies a key position in the global C cycle (Lal 2004). It is a highly heterogeneous and dynamic composite of organic molecules (such as: polysaccharides, lignin, aliphatic biopolymers, tannins, lipids, proteins and aminosugars) derived from progressively decomposed plant, animal and microbial material (Zimmermann *et al.* 2007a; Totsche *et al.* 2010).


The turnover of SOM is a balance between the inputs of material into the soil (e.g., above and
belowground litter, dissolved organic C) and the rate of SOM decomposition. The rate of
decomposition is a consequence of complex interactions and interdependence between the
organic matter and its environment. This includes: biochemical recalcitrance (compound
chemistry), physical protection (adsorption of SOM to reactive surfaces of mineral particles
and the physical protection within aggregates) (Six and Jastrow 2002), climate (temperature,
water availability), soil acidity, soil redox state (Raich and Schlesinger 1992; Kirschbaum 1995;
Stockmann *et al.* 2013) and, functional composition of the soil microbial community (Fierer
2007; Allison 2012). More recently, it has been considered that C stability is mainly dependent
on its biotic and abiotic environment, rather than the molecular structures of C inputs
(Schmidt *et al.* 2011).

In order to understand soil C dynamics, a variety of measureable C pools have been identified
within SOM according to biological stability, decomposition rate and turnover time (Krull,
Baldock and Skjemstad 2003; Trumbore 2009; Stockmann *et al.* 2013). Specifically, SOM can
be classified into three significant pools: active, resistant and inert (Trumbore 1993; Bol *et al.*
2009). The active (also termed the labile) pool contains a high C concentration and is
composed of physically available and chemically mineralizable plant material (sugars and
amino acids) (Zou *et al.* 2005; Petrokofsky *et al.* 2012). Consequently, it is less stable and plays
an essential role in the short-term nutrient cycles, with a turnover ranging from days to a few
years (Wander 2004).

The resistant pools (also known as intermediate, slow, recalcitrant or refractory) (Krull,
Baldock and Skjemstad 2003) contain physically and chemically transformed material residing
on and within the surface of clay and silt minerals. The combination of physically protected
and biochemically recalcitrant SOM (alkyl and lignin-derived aromatic C) (Coleman and
Jenkinson 1996; Petrokofsky *et al.* 2012) causes this C pool to have a turnover on decadal
timescales (Six *et al.* 2002). This pool is important for long-term C sequestration, sorption,
cation exchange capacity and soil water-holding capacity (Wander 2004).


**Commented [OV3]:** Changed from 'recalcitrant' to 'resistant' to keep consistency throughout the text

The inert (or, passive) pool has a turnover time of decades to millenia, and is central to the
stabilization of humus and soil aggregation. This pool contains highly carbonized organic
material that is resistant to microbial mineralisation (Brodowski *et al.* 2006), as well as
charcoal (i.e. black C), and is considered to have a recalcitrant structure due to its high degree
of aromaticity (Derenne and Largeau 2001). Although this pool has a low C concentration, it
can form the largest soil C fraction, especially in fine-textured tropical soils (Marin-Spiotta *et*
*al.* 2009), and can be unaffected by land-management or climate, making it the most stable
and relevant for long-term C storage (Falloon and Smith 2000).

**Commented [OV4]:** Paragraph altered to improve flow of text

Land-use change and land management studies have found that even when the bulk soil C
does not appear to be affected, the distribution of SOM pools may change due to their
differing sensitivities to environmental forcing or external perturbation (Zimmermann *et al.*
2007b; Marin-Spiotta *et al.* 2009). It is commonly accepted that the labile pools are the most
sensitive to changes in vegetation management and are identified as an indicator of soil
quality changes in the short-term (Kennedy and Papendick 1995; Islam and Weil 2000).
However, while several studies have found the labile pool to be more sensitive to land
management (Conant *et al.* 2011; Wang and Wang 2011), others have found no discernible
effect on pool size (Leifeld and Kögel-Knabner 2005). For instance, labile pools can either
increase (Poeplau and Don 2013) or decrease, depending on the magnitude of C inputs (e.g.
roots, litter fall) or the level of grazing intensity (Figueiredo, Resck and Carneiro 2010).
Quantification of different SOM pools and how they respond to land management is
important for understanding C dynamics and their relative role in the global C cycle (Trumbore
1997; Bayer *et al.* 2001). SOM turnover models use conceptual SOM pools, but now it is
possible to substitute these pools with measurable fractions of SOC (Skjemstad *et al.* 2004;
Zimmermann *et al.* 2007b). Identification and separation of these SOC pools has led to many
methods of soil fractionation, including: physical (size, density, aggregation) and chemical
(solubility, mineralogy). Density fractionation has been very successful at assessing the short
and long-term dynamics of soil C storage (Christensen 2001; Marín-Spiotta *et al.* 2008; Marin-
Spiotta *et al.* 2009; Mueller and Koegel-Knabner 2009). This procedure is based on the
application of several disaggregating treatments, dispersion, followed by density separations
using organic solutions or inorganic salts (von Lützow *et al.* 2007) and represents a variety of

pools that are related to microbial function based on the location within the soil matrix and degree of association with minerals (Krull, Baldock and Skjemstad 2003; Trumbore 2009). Six et al., 2002 used sodium polytungstate (SPT) to isolate light and heavy fractions of SOM because of its high viscosity at high concentrations. This method was later adapted by (Marin-Spiotta *et al.* 2009) and (Mueller and Koegel-Knabner 2009) to separate SOM pools into three distinct fractions: the free light fraction (active pool), occluded light fraction (resistant pool) and heavy fraction (inert pool).

The aim of this study is to gain further mechanistic insights into the impact of land-use management on soil C losses and different SOM fractions in Peruvian montane grasslands. In order to investigate the effects of burning and grazing on soil C stocks, we took advantage of an ongoing burning/grazing study that was established in July-August 2010 (Oliveras *et al.* 2014b). The specific objectives of this study were to:

a. Quantify and compare the effect of fire history and grazing on total SOC stocks and the three main SOM pools (free light fraction, occluded light fraction and heavy fraction) at different soil depths down to 30 cm;
b. Quantify differences in soil respiration and decomposition rates on historically burnt and grazed sites;
c. Evaluate the role of soil temperature and soil moisture in regulating soil respiration.

## 4. Material and methods

### 4.1 Site descriptions

The undulating terrain in the Peruvian montane grassland is commonly used by the local communities for extensive cattle grazing and although the study area is in the National Park, burning and grazing still occasionally takes place. This study included two sites that were identified as being burnt in 2003 (Wayqecha) and 2005 (Acjanaco) (Fig 1). The site at Wayqecha is located at approximately 3085 m a.s.l. in Wayqecha Biological Station (13°18′S, 71°58′W), where the mean annual precipitation is 1560 mm and mean annual air temperature is 11.8 °C. The site at Acjanaco (13°17′S, 71°63′W), is located on the Manu national park boarder at 3400 m a.s.l and has a mean annual precipitation of 760 mm and

mean annual air temperature 6.8 °C (Girardin *et al.* 2010) (Table 2). The wet season runs from October to March and there are more noticeable variations in diurnal temperatures than seasonal differences (Zimmermann *et al.* 2009). Grass species composition are similar on both sites (*Calamagrostis longearistata, Scirpus rigidus and Festuca dolichophylla)* (Oliveras *et al.* 2014a)*.* The soils are classified as Umbrisols and are typically only 30 cm deep with a thick acidic organic rich A layer overlying a thin stony B/C horizons and no O horizon (Gibbon *et al.* 2010) (soil characteristics shown Table 1). The sites are predominantly on Palaeozoic (~450 Ma) meta-sedimentary mudstones (~80 %) (Carlotto *et al.* 1996)

## 4.2 Experimental design

The sites were set up in a factorial design in July-August 2010 to investigate the effects of fire (burnt, not-burnt) and grazing (grazed, not-grazed) on soil C stocks, soil C fractions and soil respiration. The two sites (Acjanaco and Wayqecha) were selected to include a burnt and unburnt area no more than 200 m apart, which were then split into two subplots (2 x 2 m); one with fencing, constructed 2 years prior to sampling, to stop cattle grazing and one left unfenced. Each site contained a factorial combination of the two treatments i.e. burnt-not grazed; burnt-grazed; not burnt-grazed; and not burnt-not grazed (Fig. 2). The fire at Acjanaco was in 2005 and before that, this area had not been burnt since the mid-70s. The more recent fire occurred in Wayqecha in 2003, and there is no information about the disturbance history before 2003.

## 4.3 Soil respiration and environmental measurements

On each plot, four permanent PVC chamber bases (diameter 20 cm, height 10 cm) were deployed randomly for the measurement of soil surface $CO_2$ fluxes, which took place morning and afternoon at two monthly intervals from July 2011 to July 2012. Soil respiration measurements were quantified using a static flux chamber technique with a Vaisala CARBOCAP® carbon dioxide probe and temperature sensor fitted inside a PVC cylindrical chamber (diameter 20 cm, height 20 cm), covered with a gas tight lid. The rate of $CO_2$ accumulation was measured every 30 seconds for 3 minutes by placing the chamber on the fixed chamber base with a gas tight rubber seal. Simultaneously, air temperature and

atmospheric pressure were measured, using a type K thermocouple (Omega Engineering Ltd.,
UK) and Garmin GPSmap 60CSx (Garmin Ltd., USA).
Flux rates were calculated in R 3.0.2 (R_Core_Team, 2012) using the *HMR* package (Pedersen,
Petersen and Schelde 2010) by plotting the headspace concentration (ppm) against time
(minutes) for each collar, which gave a linear or non-linear regression, depending on the best
fit.
In addition, soil temperature (at 5 cm and 10 cm depth) and soil moisture (at 10 cm depth)
were simultaneously measured in three locations adjacent to the collars using a ML2x
ThetaProbe equipped with 12 cm rods (Delta-T Ltd., UK) and type K thermocouples (Omega
Engineering Ltd., Manchester, UK).
**4.4 Soil sampling and analysis**
*Soil sampling:* 50 g soil samples were taken in July 2012 with six replicates at 0-5, 5-10, 10-20
and 20-30 cm depths on each site. In many instances, the soil depths were shallow before
reaching the bedrock, so samples were only taken at 20-30 cm where possible. Soil samples
were air-dried and sieved with a 2 mm mesh sieve before being shipped to the University of
St Andrews for all further analysis (Brown and Lugo 1982).
*Bulk density:* soil bulk density was determined by the soil core method (Klute 1986).
Undisturbed soil cores (30 cm$^3$) were taken from three soil pits at 0-10, 10-20 and 20-30 cm.
The samples were dried at 105 °C for 48 hours and bulk density was estimated as the mass of
oven-dry soil divided by the core volume.
*Soil fractionation:* Soils C fractions were separated using a method developed by (Marín-
Spiotta *et al.* 2008) and (Mueller and Koegel-Knabner 2009). This method is useful for
separating SOM based on the location within the soil matrix and the degree of association
with minerals. Prior to the experiment, a sub-sample of soil was taken for moisture correction.
The air-dried soil material (15 g) was sieved in a 2mm mesh sieve to remove any living roots
and larger organic material and was then saturated with 60 mL sodium polytungstate solution
(NaPT, Na$_6$ [H$_2$W$_{12}$O$_{40}$], Sometu-Germany) at a density of 1.85g/mL and centrifuged for 45

minutes at 3600 rpm and allowed to settle overnight. The floating free light fraction (free LF) was aspirated via a pump and rinsed with 500 mL of deionised water through a 0.4 µm polycarbonate filter (Whatman Nuclepore Track Etch Membrane) to remove residual NaPT. The remaining slurry was further saturated with 60 mL sodium polytungstate solution (1.4 g $cm^{-3}$), mixed using a benchtop mixer (Mixer/Vortexer - BM1000) for 1 minute at 3200 rpm and dispersed ultrasonically (N10318 Sonix VCX500 sonicator Vibra-cell ultrasonic processor) for 3 min at 70 % pulse for a total input of 200 J/mL. Centrifugation (45 minutes at 3600 rpm) was used to separate the occluded light fraction (occluded LF) from the mineral residue and allowed to sit overnight to achieve further separation by flotation of organic debris and settling of clay particles in solution. The occluded LF was then aspirated via a pump and rinsed. In order to remove the NaPT from the heavy fraction (heavy F), deionised water was mixed with the material and centrifuged for 15 minutes at 4000 rpm 5 times. All fractions were oven dried at 100 °C overnight, weighed and physically ground to a fine powder before C analysis and isotope analysis. The recovery of the soil C density fractions was 96 %.

*Carbon analysis:* bulk soils were ground and homogenised using a grinding mill (Planetary Mono Mill PULVERISETTE) in preparation for C analysis at the University of St Andrews laboratories using a Finnegan Delta plus XP gas source mass spectrometer coupled to an elemental analyser (EA-IRMS).

*Decomposition estimates:* A decomposition experiment was set up as an additional estimate of soil organic matter mineralisation, using birch wood sticks as a common substrate. Five sticks were placed in a mesh bag with three 2 cm holes cut into each bag to allow accessibility for both microfauna and fauna. In July 2011, eighteen bags were buried at 10 cm depth in groups of six, in close proximity on each plot (Fig. 2). Three bags, one from each group, was collected every two months. The sticks were weighed before the experiment started and again after collection, once they were air dried, to determine mass loss. The rate of decomposition was then calculated from the slope of a linear regression with time against mass loss.

**4.5 Statistical analysis**

Statistical analyses were conducted in R version 3.0.2 (R_Core_Team, 2012). Outliers were observed by visual inspection of the boxplots where points outside of the hinges (third quartile) were removed and the data were checked for normal distributions. The $CO_2$ flux and volumetric water content (VWC) data were not normally distributed and therefore log transformed prior to parametric statistical analysis. Linear mixed effect models were conducted to identify any relationships between the environmental variables and soil characteristics with soil $CO_2$ fluxes for each site, individually. In this respect, mixed model restricted maximum likelihood analysis (REML) were computed using the *lme4* package (Bates *et al.* 2014) to include random intercepts for each collar and for the effect of grazing nested within the burnt sites. Analysis of variance (ANOVA) and Tukey's Honest Significant Different (HSD) post hoc test were used to examine statistically significant differences between means of the environmental data among the sites. Linear regression analysis was used on the decomposition data and tested to identify any relationships with the soil $CO_2$ fluxes. Differences in soil C between the areas were analaysed using a one-way ANOVA and Tukey's HSD post-hoc test, after testing for normality and homogeneity of variances.

**5. Results**

**5.1 Soil respiration and environmental drivers**

The overall annual $CO_2$ mean for the pooled data set, including all types of land management, was $1.04 \pm 0.04$ µmol m$^{-2}$ s$^{-1}$. The combination of grazing and burning significantly increased soil $CO_2$ fluxes at Wayqecha (2003) but not at Acjanaco (Fig 2). Regardless of land use, the plots at Wayqecha (2003) had greater variability and overall higher mean annual soil temperature (15 °C) and $CO_2$ flux ($1.30 \pm 0.08$ µmol m$^{-2}$ s$^{-1}$) compared to the sites in Acjanaco (2005) (12 °C and $0.79 \pm 0.03$ µmol m$^{-2}$ s$^{-1}$) (Table 2). The highest measured temperatures and $CO_2$ fluxes at Wayqecha were synchronously recorded during July-11, November-12 and March-12, whereas at Acjanaco the changes in $CO_2$ flux with season and temperature were less pronounced.

Season (which run from October to March), soil and air temperature were the main drivers of soil respiration (*p*-values = 0.031, 9.3 x 10$^{-7}$ and 0.0001, respectively), with higher

temperatures having a positive effect on soil $CO_2$ fluxes. However, when analyzing the grazed-
burnt plots at both Wayqecha and Acjanaco, there was no relationship between $CO_2$ fluxes
and temperature or any of the other environmental variables measured.

**5.2 Decomposition rates**
The decomposition of the birch wood sticks was slow, with an overall average weight loss of
~ 20 % in one year. Grazing alone appeared to slightly increase the rate of decomposition
when all the data were pooled together (grazed: y = 104.53 + -4.23x, $R^2$ = 0.98, not grazed: y
= 103.63 + -3.11, $R^2$ 0.94), but burning alone did not affect decomposition rate (burnt: y =
103.34 + -3.57, $R^2$ = 0.96, not burnt: y = 104.82 + -3.76x, $R^2$ = 0.97) (Fig 3). Site-specific
differences were observed for decomposition rates; for example, decomposition was
generally faster at Wayqecha compared to Acjanaco. In particular, the grazed - not burnt plot
at Wayqecha showed the fastest overall rate of decomposition (y = 101.98 + -0.19x, $R^2$ = 0.77)
and the not grazed - not burnt plots (controls) had the slowest decomposition rates (Fig 3) on
both sites.

Decomposition was not a strong overall predictor for $CO_2$ fluxes for the pooled dataset,
although there were some strong correlations between these two variables at specific study
sites. For example, there was a strong relationship between decomposition and soil $CO_2$ fluxes
at Acjanaco (y = 0.38 + -0.18x, $R^2$ = 0.99) (i.e. faster mass loss = higher soil respiration),
whereas at Wayqecha, this relationship was weak (y = 1.56 + 0.06x, $R^2$ = 0.07). Land-use did
not appear to influence the decomposition rate-soil $CO_2$ flux relationship.


**5.3 Soil C stocks**
Grazing, burning and the combination of burning and grazing did not significantly alter total
soil C at any depth down to 30 cm on either of the sites (Table 3). The overall sum of all the
measured depths showed signs of a decrease in C stocks on the grazed soils, from 189 ± 32
Mg C ha$^{-1}$ on the undisturbed sites to 130 ± 20 Mg C ha$^{-1}$ on the grazed-burnt sites, but this
was not statistically significant at the $P$ < 0.05 level. On average, Acjanaco (2003) had higher
C stocks (175 ± 17 Mg C ha$^{-1}$) compared to Wayqecha (2005) (150 ± 15 Mg C ha$^{-1}$).

The pooled dataset demonstrated that these soils have a notably large free LF (~20 %). When
looking at the different treatments and averaging the data across the soil profile (0-30 cm),
burning and grazing significantly reduced the proportion of C in the free LF (Table 4). The free
LF in the control soils made 20 % of the bulk soil mass and 30 % of the soil C content compared
to the burnt-grazed soils, which had the smallest recovery of free LF (10 %) and had
significantly lower C content (14 %). However, when analysing the depths individually, there
was only a significant loss of C in the free LF at 0-5 and 10-20 cm depth, with a reduction of ~
16 % (Fig 4). When analysing the two sites separately, the burnt- grazed soils at Wayqecha
had a significantly smaller proportion of C in the free LF at 0-5 cm ($p$-value = 0.002), whereas
at Acjanaco there were no significant differences among the land uses.

The occluded LF increased significantly after burning. The burnt soils had a much higher
proportion of C in the occluded LF (12 %) compared to the not-burnt soils (7 %), when the
data were pooled from across different soil depths (0-30 cm) for the two sites. When the data
were disaggregated by site, we found that burning produced site-specific effects on the
occluded LF in different soil depths. For example, at Wayqecha, the proportion of C in the
occluded LF was higher on the burnt soils when looking at the whole profile (0-30 cm), but
there were no significant differences among the separate soil layers. In contrast, at Acjanaco,
the burnt soils had a significantly higher proportion of C in the occluded LF in the upper most
layer (0-5 cm) (Grazed - burnt 24.7 ± 7.1 and Not Grazed - Burnt 23.5 ± 9.5 %) compared to
non-burnt soils (Grazed – not burnt = 10.3 ± 4.6 and Not grazed – not burnt = 6.1 ± 2.4 %).

The largest mass of soil C recovered was in the heavy F (~ 70 %). Overall, there were no
significant differences among the treatments when the data were pooled for the two sites
and across all four depths (0-30 cm). However, when the data were disaggregated by site, we
found site-specific treatment effects on the heavy F that varied depending on soil depth. For
example, in Wayqecha, the grazed – burnt plot had a significantly higher portion of C in the
heavy F in the 0-5 cm soil layer (67.7 ± 4.8 %) compared to the control plot (48.6 ± 3.7 %). We
also observed a significantly lower proportion of C in the grazed – not burnt plot (19.2 ± 12.2
%) in the 0-5 cm soil layer compared to the other treatments. In contrast, at Acjanaco, there
were no significant differences measured among the treatments.

**6. Discussion**

**6.1 Soil respiration and decomposition rates**

In this study, soil $CO_2$ fluxes ranged from 0.72 to 1.88 µmol m$^{-2}$ s$^{-1}$ (2.73 to 7.14 Mg C ha$^{-1}$ yr$^{-1}$)*, which is in the lower range (0.7 – 14.8 Mg C ha$^{-1}$ yr$^{-1}$) of other high elevation montane grassland studies (Cao *et al.* 2004; Geng *et al.* 2012; Muñoz, Faz and Zornoza 2013; Fu *et al.* 2014) and corroborates prior work by Oliveras *et al.*, 2014 (3.4 - 3.7 Mg C ha$^{-1}$ yr$^{-1}$). The absence of a seasonal trend in temperature and moisture has also been noted in other studies from the same region (Girardin *et al.* 2010; Teh *et al.* 2014). *The units have been converted to enable comparison to other studies but due to limited frequency of sampling in this study, this may not be sufficient data to reliably provide yearly emissions.

**Commented [OV7]:** Units changed to umol m-2 s-2 and an explanation for retaining the Mg C ha-1 yr-1 units.

Higher soil respiration and faster decomposition rates were consistently measured on the plots at Wayqecha (burnt in 2003) than at Acjanaco (2005), which is in keeping with Oliveras *et al.,* 2014. These site-specific differences may not be a reflection of the age of burning but rather Acjanaco being at a slightly higher elevation and on average 4 °C cooler. Despite the variance in mean annual temperature, the two sites both showed a positive correlation between temperature and soil respiration. Interestingly though, the decomposition rates at Acjanaco correlated with the $CO_2$ fluxes, suggesting that decay was a good predictor of $CO_2$ flux. This was in contrast to the lower elevation site in Wayqecha, where $CO_2$ fluxes did not correlate with decomposition rates, implying that autotrophic respiration or other environmental factors may have had a stronger influence on soil respiration.

Burning alone or grazing alone enhanced soil respiration and decomposition rates when these land management practices were considered separately, with soil temperature identified as the main environmental driver in each of these treatment types. However, when plots had been exposed to both burning and grazing together, soil temperature no longer correlated well with soil respiration. The combination of burning and grazing also produced higher soil respiration rates than the two treatments independently. While this pattern has been identified before in other studies (Ward *et al.* 2007), the drivers of this increase are less well understood, and the influence of grazing and burning have been known to have confounding effects (Michelsen *et al.* 2004). One potential explanation is that burning and grazing together act synergistically, and may obscure the influence of temperature due to the action of other

complex processes or drivers, such as changes in plant C allocation and autotrophic
respiration following the effects of the two combined disturbances. For example, studies have
found that when foliage is cut, photosynthate and other resources are allocated to the growth
of new shoots rather than to the roots (Schmitt, Pausch and Kuzyakov 2013), causing a decline
in root respiration (García-Oliva, Sanford and Kelly 1999). The resulting root death may
enhances heterotrophic microbial activity, counteracting the effects of reduced root
respiration.
Alternatively, burning can cause significant losses of N due to combustion, and grasses may
compensate for increased N limitation by increasing their allocation to roots, thereby
increasing root respiration and potentially promoting enhanced belowground C cycling
(Johnson and Matchett 2001). Some evidence was found for this type of response in prior
work; Oliveras *et al.,* 2014, found higher below and above-ground C stocks in undisturbed
soils. While overall net primary productivity (NPP) was higher on undisturbed sites, NPP
belowground was greater with grazing and fire, suggesting a shift in plant allocation patterns
after these disturbances.
**6.2 Belowground C stocks**
Overall, large total SOC stocks were measured in these montane grasslands (123 – 238 Mg C
ha$^{-1}$), which is in keeping with other high elevation grassland studies and are probably
attributable to low temperatures and wet conditions causing slow mineralisation of SOM and
turnover rates. For example, in the Qinghai-Tibetan Plateau grasslands and páramo
grasslands of the Colombian, Ecuadorian and Peruvian Andes, total SOC stocks can range
between 80 – 250 Mg C ha$^{-1}$ (Hofstede 1995; Zimmermann *et al.* 2010; Farley *et al.* 2012; Li
*et al.* 2013; Muñoz, Faz and Zornoza 2013; Oliveras *et al.* 2014b).
Soil C stocks were higher at Acjanaco than at Wayqecha. This is in agreement with Oliveras *et*
*al.,* 2014, although the Acjanaco sites in this previous study were higher (253 compared to
175 Mg C ha$^{-1}$ reported here), perhaps reflecting within site spatial heterogeneity. There was
no significant effect of either burning or grazing but grazing had a more negative effect than
burning on the total soil C stocks. This negligible effect of burning may be a consequence of

low intensity fires, fire-resilient grasses, and potentially low fuel loads at the time of burning (Knicker 2007). Grassland fires on slopes can move very quickly, so even when intense, the transfer of heat to the soil is less damaging due to low residence times (Rollins, Cohen and Durig 1993). As a result, surface temperatures do not typically exceed 100 °C or 50 °C at 5 cm depth (Campbell *et al.* 1995), and organic matter can only be fully volatilized between 200 and 315 °C (Knicker 2007). Even if the soils were dry at the time of burning which is possible during the dry season, then belowground temperatures would rise very slowly because of the insulating properties of air-filled pores, which curtail heat transfer belowground (Neary *et al.* 1999).

Grazing on the other hand, caused a decrease in the total SOC content compared with burning, although this numerical difference was not statistically significant at the $P < 0.05$ level. One explanation is that the grazing pressure in these sites may have been below the threshold required to cause severe degradation, supporting previous studies in the Peruvian Andes, where they also found no significant effect of grazing or burning on total SOC stocks (Gibbon *et al.* 2010; Oliveras *et al.* 2014b).

In this study, the free LF was larger than in other tropical systems (30 % of total soil C). By comparison, studies in Ecuador, Brazil and Puerto Rico found the free LF ranged from only 4-12 % of total soil C content (Paul, Veldkamp and Flessa 2008; Marin-Spiotta *et al.* 2009; Potes *et al.* 2012). However, it is difficult to compare the results of this study to other tropical fractionation studies because in general, most field sites are in tropical lowland pastures where soil C stocks tend to be lower. When comparing to other high elevational studies, for example, in permafrost meadow ecosystems in the Tibetan Qinghai Province, results are similar, with the free LF making up 27 % of the total soil C stocks (Dörfer *et al.* 2013). Comparisons are also further complicated by land-management history and methodological differences. For example, in a review of 22 grassland studies, the average fraction of soil organic C in the free LF was 13.9 %, but the range was between 1.8 and 55 % (Gregorich *et al.* 2006).

Overall, grazing significantly reduced the free LF. As grazing is known for reducing aboveground biomass (Johnson and Matchett 2001; Gibbon *et al.* 2010), a lower

Commented [OV8]: Negative wording changed

Commented [OV9]: Additional literature added

Commented [OV10]: Negative wording removed

incorporation of detritus into the soil is not surprising and has been observed in other grazing
studies (Figueiredo, Resck and Carneiro 2010; Cao *et al.* 2013). While there was a significant
decrease in the free LF, there was no significant change to the total SOC. Due to the dynamic
nature and sensitive response of this pool to land management or land-use change, other
studies have also measured reductions in the free LF, while the total SOC content appears to
be unchanged (Leifeld and Kögel-Knabner 2005; Zimmermann *et al.* 2007b; Marín-Spiotta *et al.* 2008; Cao *et al.* 2013).
*al.* 2008; Cao *et al.* 2013).

The effects of grazing on the free LF were most noticeable when grazing and burning occurred
together, in which case, the free LF showed the most pronounced declines. This was especially
evident on one of the sites (Wayqecha) in the top soil layer (0-5 cm) and mid-soil layer (10-20
cm) and could be attributed to site specific differences, such as warmer temperatures causing
higher turnover rates. To our knowledge there are no other studies assessing the impact of
grazing and burning on soil C fractions in high altitude tropical grasslands. However, studies
focusing on burning in grasslands have found a decrease in the free LF with burning history,
due to the resulting decrease in litter inputs to the soil (Fynn, Haynes and O'Connor 2003;
Potes *et al.* 2012).

> **Commented [OV11]:** More discussion on the free LF


When measuring the soil organic pools, the long-term effects of land-use can be gained by
relatively short-term experiments because burning, in theory, could have a relatively
immediate impact on all the pools of carbon. In this study, the significant increase of the
occluded LF in the burnt soils may be the results of charcoal particles (from burning) becoming
incorporated into the occluded LF. Charcoal, because of its low density, tends to reside in the

> **Commented [OV12]:** Positive wording removed

lighter fractions (Cadisch *et al.* 1996; Glaser *et al.* 2000; Sollins *et al.* 2006), despite its
recalcitrance. Because the fires took place almost ten years ago, the charcoal may no longer
be resident the free LF but may have become occluded into soil micro-aggregates, due to its
high sorptive capacity (Knicker 2007; Qayyum *et al.* 2014). This has been observed in different
soil types, such as Haplic Luvisols and Andosols (Golchin *et al.* 1997; Brodowski *et al.* 2006),
and once incorporated into micro-aggregates, charcoal can be maintained for centuries after
fire (Zackrisson, Nilsson and Wardle 1996).

> **Commented [OV13]:** Additional literature included



**7. Conclusions**
This study highlights the complexities of how land management can affect soil C dynamics in
montane tropical grasslands. The results suggest that montane grasslands are resilient to soil
C losses under moderate intensity land use. Total C stocks appeared unaffected by burning
and grazing, although a change was observed in the distribution of soil C across different soil
C fractions, with burning leading to a significant reduction in the free LF pool and an
enhancement of the occluded LF pool. Most specifically, our study shows that land
management affected the magnitude and drivers of soil respiration and decomposition.
Individually, burning and grazing alone increased soil $CO_2$ fluxes, which was apparently driven
by shifts in soil temperature. However, the combined effect of burning and grazing together
interacted synergistically, leading to enhanced soil respiration rates, while simultaneously
obscuring the role of temperature and other environmental drivers, potentially due to
changes in patterns of plant C and N allocations.

**8. Acknowledgements**
The authors wish to thank the Manu National Park forest rangers for allowing us to use their
facilities and the field technicians for their assistance. We also thank the Amazon Basin
Conservation Association for institutional support. This material is based upon work
supported by the UK Natural Environment Council under grant joint grant references
NE/H006583, NE/H007849 and NE/H006753). This publication is a contribution from the
Scottish Alliance for Geoscience, Environment and Society (http://www.sages.ac.uk).
I.Oliveras was supported with a NERC grant NE/G006385/1.


**9. Authorship**
V. Oliver designed the study, conducted the fieldwork, statistical data analysis and wrote the
manuscript. I. Oliveras designed the study, provided supervision and contributed to writing
the manuscript. J. Kala and R. Lever conducted fieldwork and laboratory analysis. Y. A. Teh
obtained funding for the work, provided supervision for the whole study and contributed to
writing the manuscript.

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

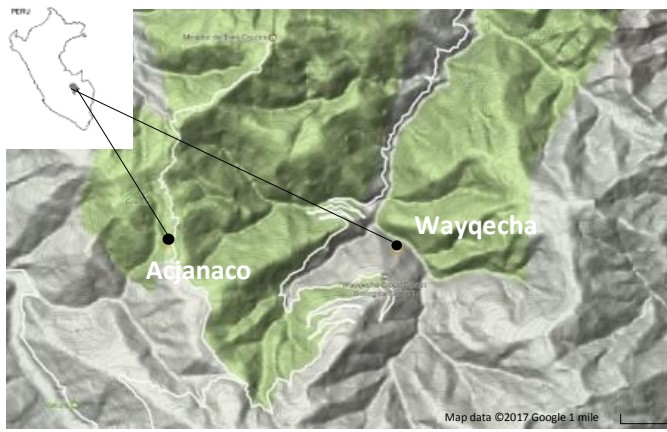

**Figure 1** Map illustrating the two sites in the high elevation montane grassland (circles).

The green area represents the Manu National Park.

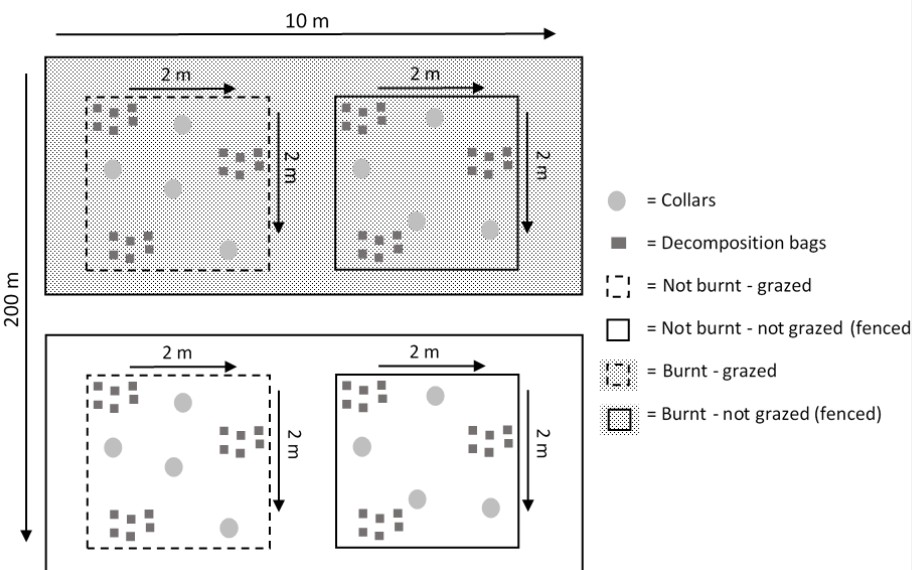

**Figure 2** Schematic diagram illustrating the set-up of the plots. This experimental design was established at both Acjanaco and Wayqecha. Soils from three pits in each plot were collected for analysis.

**Table 1** Soil description for each land management at Wayqecha and Acjanaco (mineral soil particle size taken from (Diem *et al.* 2017 - *submitted to Biogeosciences*)**.**

| Site | Land use | Bulk density (g cm$^{-3}$) | | pH | Mineral soil particle size | | |
|---|---|---|---|---|---|---|---|
| | | | | | Sand | Silt | Clay |
| | | 0-10 cm | 10-20 cm | 0-10 cm | | 0-10 cm | |
| Wayqecha (2003) | Grazed - burnt | 0.45 ± 0.03 | 0.37 ± 0.05 | 4.3 ± 0.2 | | | |
| | Not grazed - burnt | 0.25 ± 0.13 | 0.47 ± 0.03 | 4.1 ± 0.1 | | | |
| | Grazed - not burnt | 0.43 ± 0.01 | 0.61 ± 0.10 | 4.3 ± 0.1 | | | |
| | Not grazed - not burnt | 0.30 ± 0.07 | 0.46 ± 0.05 | 4.5 ± 0.2 | 43.0 ± 3.2 | 54.4 ± 3.0 | 2.6 ± 0.2 |
| Acjanaco (2005) | Grazed - burnt | 0.41 ± 0.03 | 0.47 ± 0.05 | 4.8 ± 0.2 | | | |
| | Not grazed - burnt | 0.40 ± 0.02 | 0.45 ± 0.06 | 4.4 ± 0.2 | | | |
| | Grazed - not burnt | 0.34 ± 0.03 | 0.35 ± 0.03 | 4.1 ± 0.1 | | | |
| | Not grazed - not burnt | 0.36 ± 0.06 | 0.48 ± 0.13 | 5.0 ± 0.3 | | | |

**Table 2** Annual and seasonal mean soil temperature, VWC and $CO_2$ flux for Wayqecha and Acjanaco for each land management system. Different letters down the columns represent significant differences between sites.

**Commented [OV14]:** Description of table 2 added

| Site / land use | Soil temp. (°C) at 5 cm | VWC (%) at 5 cm | $CO_2$ flux ($\mu mol\ m^{-2}\ s^{-1}$) |
|---|---|---|---|
| **Wayqecha (2003)** | 14.7 ± 0.1 | 62.3 ± 0.4 | 1.31 ± 0.09 |
| Grazed – burnt | 15.3 ± 0.3[a] | 63.4 ± 0.3[ab] | 1.88 ± 0.23[a] |
| Grazed - not burnt | 14.5 ± 0.2[ab] | 63.8 ± 0.2[ab] | 1.07 ± 0.07[b] |
| Not grazed - burnt | 14.6 ± 0.3[ab] | 60.9 ± 1.0[c] | 0.99 ± 0.08[bc] |
| Not grazed - not burnt | 14.1 ± 0.2[b] | 62.5 ± 0.8[bc] | 1.10 ± 0.07[ab] |
| Dry season | 14.1 ± 0.2 | 61.4 ± 0.8 | 1.35 ± 0.16 |
| Wet season | 15.1 ± 0.20 | 63.8 ± 0.3 | 1.31 ± 0.10 |
| Minimum | 11.6 | 29.9 | 0.22 |
| Maximum | 18 | 65.8 | 8.33 |
| **Acjanaco (2005)** | 11.6 ± 0.1 | 64. 5 ± 0.1 | 0.91 ± 0.03 |
| Grazed – burnt | 12.0 ± 0.2[c] | 64.0 ± 0.2[ab] | 0.82 ± 0.05[bc] |
| Grazed - not burnt | 11.5 ± 0.2[cd] | 64.5 ± 0.2[ab] | 0.84 ± 0.07[bc] |
| Not grazed - burnt | 11.9 ± 0.1[cd] | 64.2 ± 0.2[ab] | 0.77 ± 0.05[c] |
| Not grazed - not burnt | 10.8 ± 0.1[d] | 65.1 ± 0.2[a] | 0.72 ± 0.05[c] |
| Dry season | 11.6 ± 0.1 | 63.8 ± 0.2 | 0.81 ± 0.04 |
| Wet season | 11.7 ± 0.1 | 65.1 ± 0.1 | 0.74 ± 0.03 |
| Minimum | 9.5 | 57.1 | 0.09 |
| Maximum | 13.7 | 67.7 | 2.69 |
| GRAZED – BURNT | 13.8 ± 0.2[a] | 63.7 ± 0.2[a] | 1.35 ± 0.13[a] |
| GRAZED – NOT BURNT | 13.2 ± 0.2[a] | 64.1 ± 0.1[a] | 0.95 ± 0.05[b] |
| NOT GRAZED – BURNT | 13.3 ± 0.2[a] | 62.6 ± 0.5[a] | 0.88 ± 0.05[b] |
| NOT GRAZED – NOT BURNT | 12.6 ± 0.2[a] | 63.8 ± 0.4[a] | 0.91 ± 0.05[b] |

**Table 3**. Mean soil C content (Mg C ha$^{-1}$) for each depth and total C stocks (0-30 and 0-20 cm) on all the land management systems. Different letters down the columns within each depth represent significant differences among sites. All values are given with 1 standard error of the mean (n = 3).

| Site | Land use | Bulk C (Mg C ha$^{-1}$) | | | | Total C stock (Mg C ha$^{-1}$) | |
|---|---|---|---|---|---|---|---|
| | | 0-5 cm | 5-10 cm | 10-20 cm | 20-30 cm | 0-30 cm | 0-20 cm |
| Acjanaco | **Grazed - burnt** | 40.9 ± 6.5[a] | 31.7 ± 4.4[a] | 43.1 ± 13.4[a] | 57.6 ± | 136 ± 30[a] | 117 ± 17[a] |
| | **Not grazed - burnt** | 53.5 ± 4.5[a] | 40.9 ± 4.7[a] | 76 ± 3.7[a] | 35.4 ± | 182 ± 24[a] | 170 ± 12[a] |
| | **Grazed - not burnt** | 41.4 ± 3.2[a] | 34.7 ± 6.6[a] | 53.7 ± 16[a] | 44.2 ± | 144 ± 16[a] | 130 ± 8[a] |
| | **Not grazed - not burnt** | 40.7 ± 8.3[a] | 44.4 ± 5.4[a] | 81.4 ± 24[a] | 71.6 ± 13.4 | 238 ± 33[a] | 166 ± 22[a] |
| | **Average** | | | | | 175 ± 17[a] | 146 ± 10[a] |
| Wayqecha | **Grazed - burnt** | 40 ± 1.7[a] | 26.6 ± 1.6[a] | 40.8 ± 5[a] | 16 ± 3.2 | 123 ± 10[a] | 107 ± 8[a] |
| | **Not grazed - burnt** | 40.3 ± 3.3[a] | 16 ± 5.7[a] | 63.4 ± 21.1[a] | 44.4 ± 29.5 | 175 ± 47[a] | 131 ± 18[a] |
| | **Grazed - not burnt** | 41.3 ± 11.5[a] | 41.3 ± 9.8[a] | 42 ± 5.1[a] | 3 ± | 126 ± 24[a] | 125 ± 25[a] |
| | **Not grazed - not burnt** | 38.7 ± 5.3 | 31 ± 3.6[a] | 55.4 ± 17.3[a] | 14.8 ± 4.4 | 140 ± 31[a] | 125 ± 26[a] |
| | **Average** | | | | | 150 ± 15[a] | 122 ± 9[a] |
| Acjanaco + Wayqecha | **Grazed - burnt** | 40 ± 3[a] | 30 ± 2[a] | 42 ± 6[a] | 26 ± 11[a] | 130 ± 20[a] | 112 ± 12[a] |
| | **Not grazed - burnt** | 47 ± 4[a] | 34 ± 5[a] | 70 ± 10[a] | 42 ± 21[a] | 179 ± 36[a] | 151 ± 15[a] |
| | **Grazed - not burnt** | 41 ± 5[a] | 38 ± 5[a] | 48 ± 8[a] | 24 ± 21[a] | 135 ± 20[a] | 127 ± 16[a] |
| | **Not grazed - not burnt** | 40 ± 4[a] | 38 ± 4[a] | 68 ± 14[a] | 43 ± 14[a] | 189 ± 32[a] | 146 ± 24[a] |

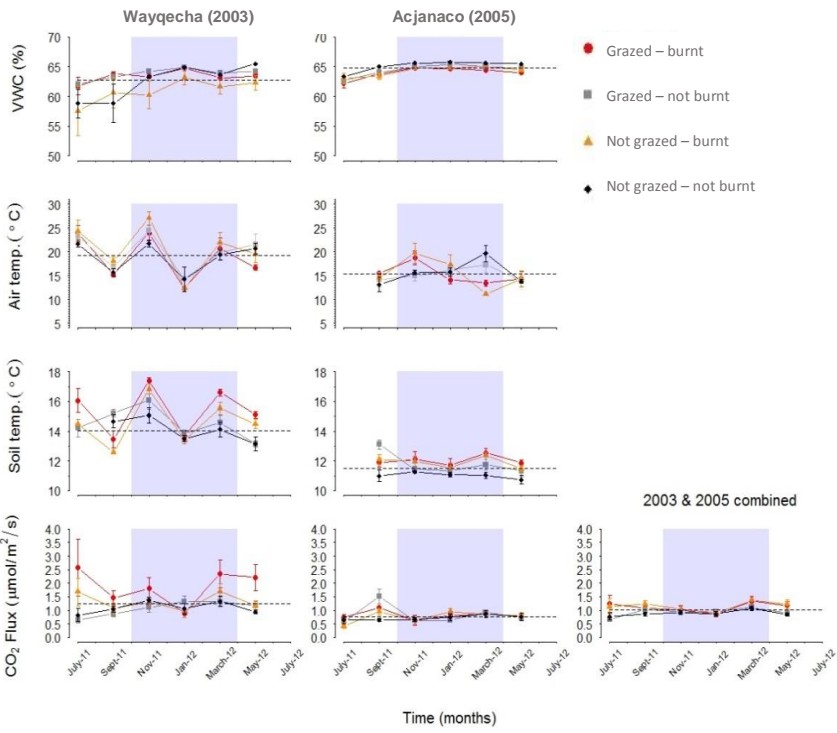

**Figure 3**. Monthly soil temperature (5 cm), air temperature, soil VWC (0-10 cm) and soil $CO_2$ flux from grazed and non-grazed subplots on sites burned in 2003 (Wayqecha) and 2005 (Acjanaco) and adjacent non burnt sites in the montane grassland. The graph on the right represents the mean $CO_2$ flux of both burnt sites combined. For $CO_2$ fluxes, each symbol is a mean of 4 chambers with morning and afternoon measurements combined and standard errors ($n$ = 8) are plotted as error bars. The dotted line represents the mean for that site and the blue band represents the wet season (Oct-March).



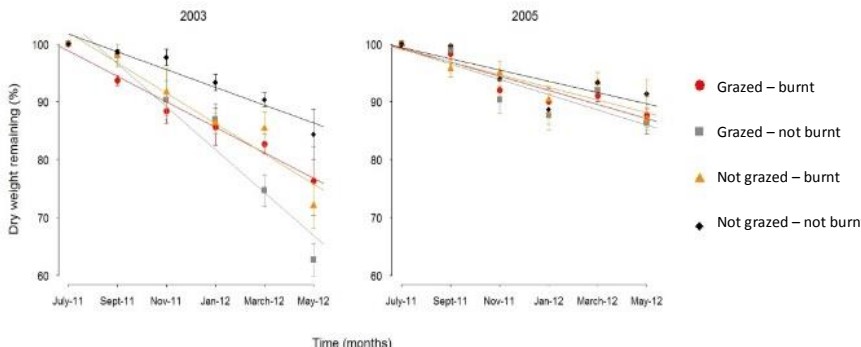

**Figure 4** Mass losses (%) of sticks from the decomposition experiment (*n* = 3) on two burnt sites (2003 = Wayqecha and 2005 = Acjanaco) with grazed subplots and control plots.


**Table 4** Mean mass recovery of density fractions and proportion of total C residing in the three density fractions (%) from the total soil profile (0-30 cm). Different letters down the columns represent significant differences.

| | Free LF | | Occluded LF | | Heavy F | |
|---|---|---|---|---|---|---|
| | Fraction of total C (%) | Mass of soil recovered (%) | Fraction of total C (%) | Mass of soil recovered (%) | Fraction of total C (%) | Mass of soil recovered (%) |
| **Grazed - burnt** | $14.0 \pm 5.3^b$ | $9.9 \pm 3.6^a$ | $10.8 \pm 2.6^{ab}$ | $9.8 \pm 3.4^{ab}$ | $76.0 \pm 8.0^a$ | $78.4 \pm 7.2^a$ |
| **Grazed - not burnt** | $22.7 \pm 13.3^{ab}$ | $16.2 \pm 8.5^a$ | $8.9 \pm 2.1^{bc}$ | $5.3 \pm 1.6^{bc}$ | $68.3 \pm 14.0^a$ | $76.7 \pm 8.1^a$ |
| **Not grazed - burnt** | $19.7 \pm 8.3^{ab}$ | $15.1 \pm 8.5^a$ | $14.2 \pm 2.5^a$ | $11.3 \pm 4.7^a$ | $66.1 \pm 10.5^a$ | $76.6 \pm 8.3^a$ |
| **Not grazed - not burnt** | $30.0 \pm 5.7^a$ | $19.5 \pm 5.5^a$ | $5.2 \pm 0.8^c$ | $4.3 \pm 0.7^c$ | $64.7 \pm 6.1^a$ | $69.7 \pm 5.8^a$ |

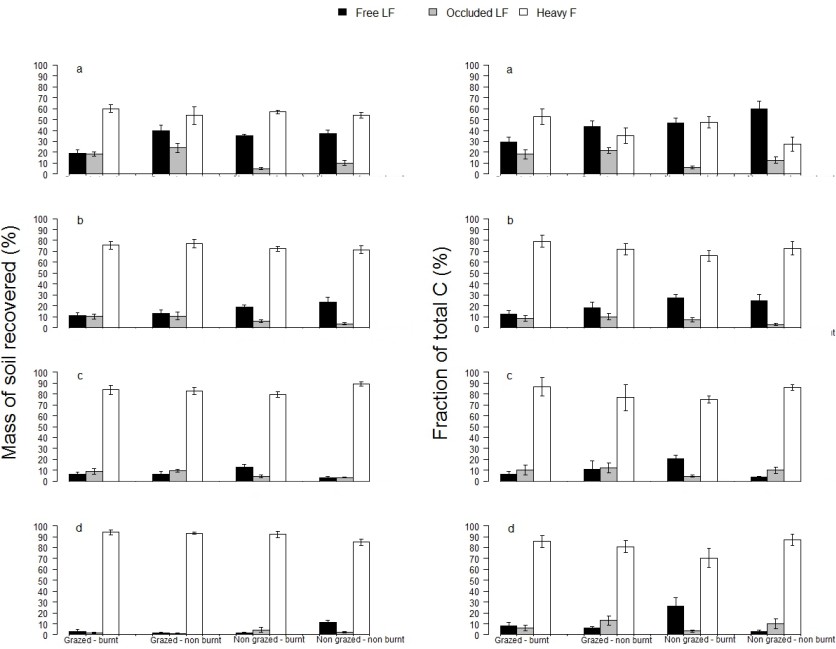

**Figure 5** Mass of soil recovered in the three density fractions (%) on the four left bar plots and the proportion of total C residing in the three density fractions (%) on the four right bar plots for the different land uses (a = 0-5 cm, b = 5-10 cm, c = 10-20 cm, d =  20-30 cm). Error bars indicate 1 standard error of the mean (*n* = 6).