# Peer review of "1. Title page The effects of burning and grazing on soil carbon dynamics in managed Peruvian tropical montane grasslands"

_Biogeosciences, 2017_

## Referee Comment (RC1) · Anonymous Referee #1 · 9 May 2017

General Comments:

The study aims to investigate grazing and burning effects on montane tropical grassland soils. Soil carbon dynamic was investigated on two different sites with four combination of land management by gaining labile and stabile fractions along with decomposition measurement and soil C fluxes. Due to the complex design (4 management combination at only two sites, different sampling depth) and different used methods, it is challenging to work out clear effects of the management system on soil C dynamic and its long/short-term implications.

One critical point may be that differences in carbon stocks at Ajanaco found by this study (170? Mg C ha-1) and (253 Mg C ha-1) by an previous study (Oliveras et al.

2014) are substantial (larger than differences between management systems). The authors related the difference to spatial heterogeneity (L.383). If there is such a high variability, how can differences related to management differences?

The work is based on a concept of different soil organic matter pools and stability. However, it is not stated which separated soil fraction correspond to which pool and stability. Therefore I cannot understand how the authors can make a statement on the effect of long-term stability of the different management systems. Moreover, as grazing is excluded only for one year before experiments started? Burning took place 6-8 years before soil sampling and grazing activity was excluded one year (?) before sampling and measurements? One main finding – as stated in the abstract (.L49-51)- is that long-term C storage on occluded LF and HF is not impacted. What did you mean by long-term? One year? After the concept the occluded LF has a slower turnover compared to the free LF. Consequently effects of grazing may be not visible after one or two years in the occluded fraction. Or if so, what does this implicate for soil carbon dynamics? If the proportion of recalcitrant soil C increases after burning in the occluded, what are the consequences for long-term storage? Does burning favour Csequestration? Can be long-term effects gained by relatively short-term experiments? In this sense, please check title and the discussion section.

The cited literature could be improved: New literature and concepts about stability of SOC could improve the manuscript, such as Schmidt et al. 2011, Nature 478, 49-56. In addition, a literature overview about density fractions is missing. E.g. one tropical study is cited for many tropical, temperate and boreal studies. L.118. The same citation is used for a generally ranking of the results. L.-401-402. Including literature about density fraction and turnover times could improve the manuscript. On the other hand general statements (management history; L.60-61) are documented with 5 citations.

**Specific comments:**

L.40-42. I would suggest including only percentage of soil C and not bulk soil to improve

readability.

L.46-47: As autotropic respiration was not measured, I would omit these speculations in the abstract.

L. 49-51: Please specify what you mean by long-term

L.58-65: How often are these grasslands burnt? Every 10-20 years, once for pasture establishment? How important is burning for these systems?

L.69: What do you mean by soil C balances?

L.92-101: see general comment on new literature on SOC stability and ecosystem properties (e.g. Schmidt et al. 2011)

L.124/L: Which particle-sizes were separated? Where are the results?

L.133: please specify different management systems

L.133-134: please specify labile and stable OM pools

L.135-137: Which environmental drivers do you mean except soil temperature and VWC? Please specify the objective

Table 1: I would like to have the given information (BD, pH C:N, Soil C) at least for both sites and different depth (and management system). For me it is not clear which soil is descripted in Table 1.

L.163-164. please add information: How long were these sites were grazed / not grazed and about fire frequency.

L.185: How were the bi-monthly measurements extrapolated to gain annual emissions? What is the uncertainty of the annual emission? The annual emission is only based on 6 measurement days – without information on soil temperature course of the year. Soil respiration is driven by soil temperature (L270), but measurements only included day measurements at a very low frequency. What do you want to express with the annual

СЗ

emission rates?

L.203-208: Does the free LF included (living) roots or were they sorted out before? (This would have major implications for the yield of free LF), see also comment L.299

L.203-223: I am missing information about soil C recovery in density fractions: bulk soil measured = 100%, sum of soil C in density fractions = ?%

L.261-262: Does at Acjanaco grazing and burning significantly increase soil CO2 fluxes? From Figure 2, I do not get the impression.

L.269: How is season defined? By soil temperature and VWC? Are soil temperature and air temperature not strongly correlated?

L. 299: belowground carbon stock = soil carbon stock + living roots?

L.300-305: and L381-383. Comparison of soil carbon stocks of Acjanco from different studies Oliveras et al. 2017 submitted and Oliveras et al. 2014): If there is a high spatial variability (170 vs 253) how can be differences found at the different sites (grazed-ungrazed-burned-not burned) traced back to differences in management and not also to spatial variability? Please check carbon stock 152 vs 170. I have difficulties to account the number of replicates of soil C sampling (from design description I got the impression of 4 replicates, Table 2 : n=3. Eventually a small graphic with sampling design would help to understand the experimental design.

L. 362. As heterotrophic respiration is not measured: may enhances.., as it is a speculation

L 364: Is the N loss reflected by different C/N ration in soil?

L.376: It would be nice to have a range of soil C stocks found in montane grassland soils

L.385-L399. There was no effect of burning on total soil C and no significant effect of grazing on total soil C. However grazing had a more negative effect on total soil C.

please clarify.

L.401-411: Please expand literature and discussion. In addition, please check the number cited (10%) and carefully consider the land use type. I do not understand L 403-404. It would be nice to have the range of free LF found in tropical soils in order to rank and interpret the gained results (L 403-406).

L.413-420: Does this mean that burning favours long-term stabilisation of soil C as charcoal? It is it is stated (L49-51), that the long term storage in the occluded fraction was not negatively impacted, but has a positive effect?

---

## Referee Comment (RC2) · Anonymous Referee #2 · 16 May 2017

In the present study Oliver and colleagues tried to identify impacts of land management practices (i.e. grazing and burning) on soil C dynamics in the top 30 cm of Peruvian montane grassland soils. Soil and gas sampling was conducted at two different sites 8/9 years and 6/7 years, respectively, after burning. Oliver et al. point out that both sites differed in elevation, mean annual precipitation and air temperature but were similar in soil conditions and grass species composition. By means of density and particle-size fractionation methods they quantified soil C content in free light, occluded and heavy fractions. Soil CO2 efflux and decomposition rate measurements were additionally conducted using static flux chamber technique and mesh bags, respectively. Overall, Oliver et al. observed an increase of soil CO2 efflux and decomposition rates and a

decrease of the soil C proportion in the free light fraction in the grazed+burned plots. Total soil C stocks did not change.

It is an interesting study but my major concern is about the experimental design. There is no random plot or site selection. Hence, there is no true replicate in the whole study. This makes it very difficult or even impossible to interpret the results in an appropriate way. Unfortunately, results are mainly analysed/described based on pooled data (P9, L260-261; P10, L280-284, L292, L308-315; P11, L319-324;) derived from two different sites with significant site-specific differences and differences in fire history (e.g. P10,L284; P11,L338; P5,L145). Then, this information even gets lost throughout discussion and conclusions (e.g. P12, L347-352; P13, L385-386, L395-396; P14, L413-420).

Specific comments: P1,L3: Title is too general.

P1,L32: …impacts of burning but not of fire history. Oliver et al. have not studied effects of past fire frequency or intensity on soil C dynamics but rather differences in soil C dynamics at two sites 8/9 years and 6/7 years, respectively, after a burning event.

P6,L162: Explain "puna areas".

P6,L162: Do you have more information about the "unburnt" grassland area. I guess that this "control" grassland area has been burnt as well in the past. Are there potential differences between both "control" sites?

P5,L132: grazing and burnt plots.

P5,L133-134: Please explain the connection between labile and stable organic matter pools with your quantified soil C content in free light, occluded and heavy fractions more in detail! What is what?

P5, L135-L137: Please do not pool the data among sites but rather describe/interpret the site-specific patterns.

P6, L159-170: A figure presenting the spatial distribution of the plots at both sites would be great.

P8,L234: Please explain "proximity". Did the bags cover the whole area? What was the distance between buried bags?

―――――――――――――――――――――

---

## Author Comment (AC2) · 15 Jul 2017

The comment was uploaded in the form of a supplement:
https://www.biogeosciences-discuss.net/bg-2017-113/bg-2017-113-AC2-supplement.pdf

---

## Author Response (AR1)

**DETAILED RESPONSE TO REFEREES**

On behalf of my co-authors, I would like to thank the two anonymous referees for their thoughtful and constructive comments on our manuscript. A detailed description of how we have responded to the referees comments is provided below.

**RESPONSE TO REFEREE 1**

*One critical point may be that differences in carbon stocks at Acjanaco found by this study (170? Mg C ha-1) and (253 Mg C ha-1) by a previous study (Oliveras et al. 2014) are substantial (larger than differences between management systems). The authors related the difference to spatial heterogeneity (L.383). If there is such a high variability, how can differences related to management differences?*

*The work is based on a concept of different soil organic matter pools and stability. However, it is not stated which separated soil fraction correspond to which pool and stability. Therefore I cannot understand how the authors can make a statement on the effect of long-term stability of the different management systems. Moreover, as grazing is excluded only for one year before experiments started? Burning took place 6-8 years before soil sampling and grazing activity was excluded one year (?) before sampling and measurements?*

*One main finding – as stated in the abstract (.L49- 51)- is that long-term C storage on occluded LF and HF is not impacted. What did you mean by long-term? One year? After the concept the occluded LF has a slower turnover compared to the free LF. Consequently effects of grazing may be not visible after one or two years in the occluded fraction. Or if so, what does this implicate for soil carbon dynamics? If the proportion of recalcitrant soil C increases after burning in the occluded, what are the consequences for long-term storage? Does burning favour C sequestration? Can be long-term effects gained by relatively short-term experiments? In this sense, please check title and the discussion section.*

*The cited literature could be improved: New literature and concepts about stability of SOC could improve the manuscript, such as Schmidt et al. 2011, Nature 478, 49-56. In addition, a literature overview about density fractions is missing. E.g. one tropical study is cited for many tropical, temperate and boreal studies. L.118. The same citation is used for a generally ranking of the results. L.-401-402. Including literature about density fraction and turnover times could improve the manuscript. On the other hand general statements (management history; L.60-61) are documented with 5 citations.*

*1.* **Author's response:**

One source of variance in soil C stocks is due to differences in depths of the soil profile. All sites contained soil to a depth of 20 cm; however, beyond a depth of 20 cm, there was higher variability, with some plots containing soil while others contained parent material (i.e. regolith). Please see Tables 1 and 2 presented here for a breakdown of total soil C stock estimates (Table 1) and a breakdown of C stock content in each soil layer (Table 2).

In Oliveras et al (2014), all C stocks were reported to 0-30 cm, and this is the main source of disagreement in one of the sites (Acjanaco). The values reported in both studies are similar for the other site (Wayqecha). However, despite these differences in means and the heterogeneity in soil C stocks, our statistical tests still indicate significant differences due to land use. That is, even with a very remarkable variability, we controlled the significance level of the tests at 5% in order to avoid Type I and type II errors, and therefore we are confident that any statistically significant differences are due to real differences arising from land use practices and not from soil heterogeneity.

**Table 1**. Total C stocks 0-30 and 0-20 cm for Wayqecha and Acjanaco.

| Site | Land use | Total C stock 0-30 cm (Mg C ha$^{-1}$) | Total C stock 0-20 cm (Mg C ha$^{-1}$) |
|---|---|---|---|
| Acjanaco | Grazed burnt | 136 ± 30 | 117 ± 17 |
| | Non grazed burnt | 182 ± 24 | 170 ± 12 |
| | Grazed non burnt | 144 ± 16 | 130 ± 8 |
| | Non grazed non burnt | 238 ± 33 | 166 ± 22 |
| | Average | 175 ± 17 | 146 ± 10 |
| Wayqecha | Grazed burnt | 123 ± 10 | 107 ± 8 |
| | Non grazed burnt | 175 ± 47 | 131 ± 18 |
| | Grazed non burnt | 126 ± 24 | 125 ± 25 |
| | Non grazed non burnt | 140 ± 31 | 125 ± 26 |
| | Average | 150 ± 15 | 122 ± 9 |

**Table 2.** Soil organic carbon (SOC) stocks down the soil profile Wayqecha and Acjanaco.

| Land use | Depth (cm) | Acjanaco SOC (Mg C ha$^{-1}$) | Wayqecha SOC (Mg C ha$^{-1}$) |
|---|---|---|---|
| **Grazed burnt** | 0-5 | 40.91 ± 6.45 | 40.03 ± 1.66 |
| | 5-10 | 32.74 ± 4.37 | 26.64 ± 1.56 |
| | 10-20 | 43.08 ± 13.42 | 40.80 ± 4.99 |
| | 20-30 | 57.61 ± | 15.98 ± 3.18 |
| **Non grazed burnt** | 0-5 | 53.50 ± 4.54 | 40.33 ± 3.29 |
| | 5-10 | 40.93 ± 4.67 | 27.14 ± 5.73 |
| | 10-20 | 76.04 ± 3.69 | 63.35 ± 21.21 |
| | 20-30 | 35.44 ± | 44.39 ± 29.46 |
| **Grazed non burnt** | 0-5 | 41.37 ± 3.16 | 41.29 ± 11.46 |
| | 5-10 | 34.65 ± 6.59 | 41.27 ± 9.76 |
| | 10-20 | 53.74 ± 15.98 | 42.03 ± 5.13 |
| | 20-30 | 44.15 ± | 3.03 ± |
| **Non grazed non burnt** | 0-5 | 40.66 ± 8,28 | 38.70 ± 5.27 |
| | 5-10 | 44.38 ± 5.39 | 31.00 ± 3.64 |
| | 10-20 | 81.44 ± 23.98 | 55.39 ± 17.27 |
| | 20-30 | 71.59 ± 13.37 | 14.75 ± 4.43 |

All values are given with 1 standard error of the mean (n = 3). Except for at 20-30 cm for Acjanaco grazed burnt, grazed non burnt and Wayqecha grazed non burnt where only one sample could be taken.

With respect to some of the questions the referee raised with respect to soil C fractions the separated C fractions correspond to:

1) Free LF = labile pools (1 to 5 years)
2) Occluded LF = intermediate pools (+ 10 years)
3) Heavy F = stable pools (centuries to millennia)

With regard to the referees' concerns about the grazing treatments, and what constitutes "long-term"; grazing was excluded for 2 years prior to sampling and measurements. By long term, we mean that carbon with long residence times (i.e. the "heavy" or mineral-associated fraction, which turns over on the timescale of centuries to millennia, was not impacted by fire or grazing. The manuscript has since been revised to clarify these points and to include new cited literature and concepts.

- When measuring the soil organic pools, the long-term effects of land use can be gained by relatively short-term experiments because burning, in theory, could have a relatively immediate impact on all the pools of carbon. Previous studies have also shown that moderate burning can favour C sequestration by incorporating charcoal deposits in the intermediate and stable pools.

- Of course a longer term study (+10 years) would be ideal but not possible in this case study. However, the findings from this study provide the first set of data on how land use affects different soil C pools on an understudied ecosystem. This can then provide a basis for further studies.

- Grazing has been occurring for decades on these grasslands and although the study only prevented grazing for 2 years, the soils are in a continuous dynamic state. Therefore, even though the full affect will not be seen in the more stable pools, it is interesting to see how all the pools are responding to recovery.

We thank the referee for his/her suggestion to change the title of the manuscript to provide greater specificity; we will consider altering the title for the revised version of the text.

**Author's response:**

*2. L.40-42. I would suggest including only percentage of soil C and not bulk soil to improve readability.*

**Authors' response:** "20 % of bulk soil is correct" but for clarity, the sentence has been edited to read: "20 % of the material was recovered in the free LF".

*3. L.46-47: As autotropic respiration was not measured, I would omit these speculations in the abstract.*

**Authors' response:** *Autotrophic respiration has been omitted.*

*4. L. 49-51: Please specify what you mean by long-term*

**Authors' response:** ~ 10 + years.

*5. L.58-65: How often are these grasslands burnt? Every 10-20 years, once for pasture establishment? How important is burning for these systems?*

**Authors' response:** Manuscript has been changed to include more details about burning in this region:

"Every year, especially in the dry season, large areas of these grasslands are burned to support traditional cattle grazing, which has been apparent since the early 1500s (Ellenberg, 1979; Balslev and Luteyn, 1992; Molinillo and Monasterio, 1997). Fires for agricultural clearing and maintenance of these highly productive forage grasses is of considerable importance in these ecosystems and for the livelihood of the local people (Ellenberg 1958; Janzen 1973; Balser and Wixon, 2009). Evidence of fire scars and charcoal deposits along the forest-puna tree line demonstrate a gradual encroachment into the adjacent tropical montane cloud forest (Lægaard 1992)."

**6.** *L.69: What do you mean by soil C balances?*
**Authors' response:** C balances has been changed to C dynamics.

**7.** *L.92-101: see general comment on new literature on SOC stability and ecosystem properties (e.g. Schmidt et al. 2011)*
**Authors' response:** We thank the referee for the suggested reference. The manuscript has now been improved with a detailed literature review including more information about density fractions and turnover times.

**8.** *L.124/L: Which particle-sizes were separated? Where are the results?*
**Authors' response:** Particle-size has been omitted. The method only included density fractionation.

**9.** *L.133: please specify different management systems*
**10.** *L.133-134: please specify labile and stable OM pools*
**Authors' response:** Manuscript changed to: "Evaluate the effect of fire history and grazing on the free LF, occluded LF and heavy F soil carbon pools"

**11.** *L.135-137: Which environmental drivers do you mean except soil temperature and VWC? Please specify the objective*
**Authors' response:** Manuscript changed to: "Quantify differences in soil respiration and evaluate the role of soil temperature and soil moisture in regulating soil respiration."

***12.*** *Table 1: I would like to have the given information (BD, pH C:N, Soil C ) at least for both sites and different depth (and management system). For me it is not clear which soil is described in Table 1.*

**Authors' response:** A table has been included to include the soil characteristics for site and at two soil depths (0-10, 10-20 cm).

***13.*** *L.163-164. please add information: How long were these sites were grazed / not grazed and about fire frequency.*

**Authors' response:** The fire at Acjanaco was in 2005 and before that, this area had not been burnt since the mid-70s. The most recent fire occurred in Wayqecha in 2003, and we do not have information about the disturbance history before 2003. We also do not have information about the grazing history.

***14.*** *L.185: How were the bi-monthly measurements extrapolated to gain annual emissions? What is the uncertainty of the annual emission? The annual emission is only based on 6 measurement days – without information on soil temperature course of the year. Soil respiration is driven by soil temperature (L270), but measurements only included day measurements at a very low frequency. What do you want to express with the annual emission rates?*

**Authors' response:** Reviewer made a valid point and the calculations have been edited to average rather than annual emissions.

***15.*** *L.203-208: Does the free LF included (living) roots or were they sorted out before? (This would have major implications for the yield of free LF), see also comment L.299*

**Authors' response:** Methods section edited to include: "The air-dried material was sieved in a 2 mm mesh sieve to remove any living roots and larger organic material and was then saturated…."

***16.*** *L.203-223: I am missing information about soil C recovery in density fractions: bulk soil measured = 100%, sum of soil C in density fractions = ?%*

**Authors' response:** The recovery of the soil C density fractions was 96 %, which has now been included in the manuscript

**17.** *L.261-262: Does at Acjanaco grazing and burning significantly increase soil CO2 fluxes? From Figure 2, I do not get the impression.*

**Authors' response:** The reviewer is correct in stating that burning and grazing did not significantly increase $CO_2$ fluxes at Acjanaco. The sentence has been reworded to:

"However, this was only noticeable at Wayqecha (2003) and not at Acjanaco (2005) (Fig 2)."

**18.** *L.269: How is season defined? By soil temperature and VWC? Are soil temperature and air temperature not strongly correlated?*

**Authors' response:** The wet season runs from October to March, which has been cited in other studies for this region and is defined by precipitation. For the linear mixed model, season was included as a categorical variable. Correlation was checked for soil and air temperature but were not strongly correlated.

**19.** *L. 299: belowground carbon stock = soil carbon stock + living roots?*

**Authors' response:** The belowground carbon stock does not include living roots. Subheading changed to "Soil C stocks".

**20.** *L.300-305: and L381-383. Comparison of soil carbon stocks of Acjanco from different studies Oliveras et al. 2017 submitted and Oliveras et al. 2014): If there is a high spatial variability (170 vs 253) how can be differences found at the different sites (grazedungrazed-burned-not burned) traced back to differences in management and not also to spatial variability? Please check carbon stock 152 vs 170. I have difficulties to account the number of replicates of soil C sampling (from design description I got the impression of 4 replicates, Table 2 : n=3. Eventually a small graphic with sampling design would help to understand the experimental design.*

**Authors' response:** Please refer to authors' response 1. A diagram showing the sampling design has now been included.

**21.** *L. 362. As heterotrophic respiration is not measured: may enhances.., as it is a speculation*

**Authors' response:** Manuscript edited to include "May enhance…"

**22.** *L 364: Is the N loss reflected by different C/N ration in soil?*

**Authors' response:** C/N ratio wasn't mentioned in the stated study but this sentence can be taken out to avoid any confusion.

**23.** *L.376: It would be nice to have a range of soil C stocks found in montane grassland soils*

**Authors' response:** The manuscript has been edited to include a range of soil C stocks found in montane grasslands.

**24.** *L.385-L399. There was no effect of burning on total soil C and no significant effect of grazing on total soil C. However grazing had a more negative effect on total soil C. please clarify.*

**Authors' response:** The wording in the manuscript has been changed to clarify that there was no significant effect of either burning or grazing but that grazing had a more negative effect than burning on total soil C.

**25.** *L.401-411: Please expand literature and discussion. In addition, please check the number cited (10%) and carefully consider the land use type. I do not understand L 403-404. It would be nice to have the range of free LF found in tropical soils in order to rank and interpret the gained results (L 403-406).*

**Authors' response:** Manuscript has been edited to include a range of free LF found in tropical soils and a discussion with more cited literature.

**26.** *L.413-420: Does this mean that burning favours long-term stabilisation of soil C as charcoal? It is it is stated (L49-51), that the long term storage in the occluded fraction was not negatively impacted, but has a positive effect?*

**Authors' response:** Soil carbon and charcoal carbon stocks and their dynamics in the soil profile after fire is limited but some previous studies on burning have suggested that charcoal contributes to the slow carbon pools in soils, so this was one potential explanation for why we saw a positive effect in the occluded LF.

**RESPONSE TO REFEREE 2**

*27. It is an interesting study but my major concern is about the experimental design. There is no random plot or site selection.*

**Authors' response:** With all due respect to the referee, this is an incorrect interpretation of our experimental design. Treatment plots were set-up according to a randomised block design (L128) (see Oliveras et al. 2014, Table 1). We acknowledge that the manuscript may have not explained this clearly enough, and we have therefore re-written this part to clarify this point.

*28. Hence, there is no true replicate in the whole study. This makes it very difficult or even impossible to interpret the results in an appropriate way.*

**Authors' response:** The key criticism that the referee raises here is that our study is an example of a pseudo-replicated experiment, and that the study may therefore be invalid. However, we respectfully disagree with this perspective, and present three counter-arguments to this criticism here. There is a long-standing and well-established debate in the soil science and ecological literature about whether or not pseudo-replication in field experiments invalidates them (Davies and Gray, 2015;Hurlbert, 1984;Schank and Koehnle, 2009;Pennock, 2004). The consensus that has emerged from this 30-year old debate is that pseudo-replication alone does not invalidate an experiment (Davies and Gray, 2015;Hurlbert, 1984;Schank and Koehnle, 2009;Pennock, 2004).

First, provided that the experimental design allows for appropriate interspersion of experimental treatments, the problem of pseudo-replication can be ameliorated by ensuring quasi-independence of experimental treatments from each other by dispersing them in space and time (Hurlbert, 1984). We were mindful of this concern in designing our experiment, and achieved appropriate interspersion of our treatments by setting-up the experiment according to a randomised block design (Oliveras et al 2014). This is one of the approaches recommended by statisticians for ameliorating the effects of pseudo-replication (see point 27) (Hurlbert, 1984).

Second, scientists have argued that even if a study is pseudo-replicated, this does not mean, *a priori*, that these studies are invalid or fundamentally flawed. This is particularly true for experiments where practical circumstances do not allow for the implementation of a fully controlled and replicated study (Davies and Gray, 2015;Hurlbert, 1984). For example, many natural disturbances (*e.g.* fire, landslides, storm events, volcanic eruptions, pest outbreaks) are often difficult to predict and almost impossible to replicate, particularly at large spatial scales (Davies and Gray, 2015;Schank and Koehnle, 2009). Likewise, many anthropogenic disturbances (e.g. biomass burning, clear-felling, hydraulic mining, peatland drainage) may be difficult to simulate at realistically large spatial scales, due to constraints imposed by time and resources (Davies and Gray, 2015;Schank and Koehnle, 2009). For example, Davies and Gray (2015) and Schank and Koehnle (2009) assert that provide precautions are taken in the design of experiments and analysis of the data (Hurlbert, 1984), these types of disturbance- or landscape-scale experiments are still interpretable and valid. This line of argumentation is particularly salient for the work we have presented here. Because the research was conducted in Manu National Park, where burning is prohibited by park authorities, we were unable to conduct large-scale controlled burns to simulate the effects of wildfire. Therefore, our only recourse – knowing that burning is an important disturbance in these high elevation ecosystems – was to select study sites that burned naturally, accounting as far as possible for the effects of differences in key pedogenic factors (i.e. parent material, time since disturbance, relief/topography, climate, and biota). To account for differences in site age (i.e. time since burning) in our study sites, we incorporated time as a factor in our mixed effects modelling. Moreover, there is already a scientific publication about grassland productivity where this exact experimental designed was used (Oliveras et al 2014) and the reviewers at that time did not have any concerns on this matter.

Third, mindful of the potential problems posed by pseudo-replication for interpreting the data, we implemented a sampling design where we quantified key process-based variables (e.g. decomposition rate, temperature, moisture), in order to deepen our mechanistic understanding of how soil C stocks were linked to the factors that regulate their turnover and loss. We also implemented some degree of control in our management studies, by installing grazing exclosures. While these measures do not negate the issue of pseudo-replication per se, they establishes the mechanistic relationship between soil C stocks and their control variables (e.g. grazing, organic matter decay rate, temperature, moisture), enabling us to establish if disturbance (fire) and land management practices (presence or absence of cattle) were linked to underlying shifts in control variables.

**29.** *Unfortunately, results are mainly analysed/described based on pooled data (P9, L260-261; P10, L280-284, L292, L308-315; P11, L319-324;) derived from two different sites with significant site-specific differences and differences in fire history (e.g. P10,L284; P11,L338; P5,L145). Then, this information even gets lost throughout discussion and conclusions (e.g. P12, L347-352; P13, L385-386, L395-396; P14, L413-420).*

**Authors' response:** With all due respect to the referee, this is a misapprehension of how we approached our data analysis. In order to account for the potential effects arising from different aged sites, we included time since burning as a variable in our mixed effects model. Moreover, we included key environmental variables (e.g. temperature, moisture) and site properties (e.g. organic matter decay rate) as co-variates, to take into account the role of underlying site differences in modulating soil C stocks. The only reason the two sites were not discussed separately in parts of the text is because the mixed effects model indicated that there was no significant difference arising from different times since burning; however, we want to emphasise that for the statistical analyses themselves, the data were not pooled but always included time since burning as an independent variable. The revised manuscript will be altered to better clarify this point.

*P1,L3: Title is too general.*
**Authors' response:** We thank the referee for his/her suggestion to change the title of the manuscript to provide greater specificity; we will consider altering the title for the revised version of the text.

**30.** *P1,L32: . . .impacts of burning but not of fire history. Oliver et al. have not studied effects of past fire frequency or intensity on soil C dynamics but rather differences in soil C dynamics at two sites 8/9 years and 6/7 years, respectively, after a burning event.*

**Authors' response:** Manuscript has been changed to "impacts of burning…"

**31.** *P6,L162: Explain "puna areas".*

**Authors' response:** Changed to: *"Both puna sites were selected".*

**32.** *P6,L162: Do you have more information about the "unburnt" grassland area. I guess that this "control" grassland area has been burnt as well in the past. Are there potential differences between both "control" sites?*

**Authors' response:** We do not have information on the "unburnt" areas. Only that they have not been burnt since the late 70s. Potentially there are differences between the "control" sites in their burning history.

**33.** *P5,L132: grazing and burnt plots.*

**Authors' response:** Text has been changes to "grazing and burnt plots".

**34.** *P5,L133-134: Please explain the connection between labile and stable organic matter pools with your quantified soil C content in free light, occluded and heavy fractions more in detail! What is what?*

**Authors' response:** Labile pool = freeLF, Stable pools = occluded LF and heavy F. A more detailed literature review on soil fractionation has been included in the manuscript.

**35.** *P5, L135-L137: Please do not pool the data among sites but rather describe/interpret the site-specific patterns.*

**Authors' response:** A description of each site has now been included as well as the pooled data.

**36.** *P6, L159-170: A figure presenting the spatial distribution of the plots at both sites would be great.*

**Authors' response:** A figure showing the spatial distribution of the plots has been included.

**37.** *P8,L234: Please explain "proximity". Did the bags cover the whole area? What was the distance between buried bags?*

**Authors' response:** The decomposition experiment was done in triplicate on each plot, with 6 bags buried no more than 30 cm apart for each experiment. The 3 decomposition experiments were randomly located within each plot to cover the heterogeneity on the plot.

Balser, T. C., and Wixon, D. L.: Investigating biological control over soil carbon temperature sensitivity, Global Change Biology, 15, 2935-2949, 10.1111/j.1365-2486.2009.01946.x, 2009.

Davies, G. M., and Gray, A.: Don't let spurious accusations of pseudoreplication limit our ability to learn from natural experiments (and other messy kinds of ecological monitoring), Ecology and Evolution, 5, 5295-5304, 10.1002/ece3.1782, 2015.

Hurlbert, S. H.: Pseudoreplication and the Design of Ecological Field Experiments, Ecological Monographs, 54, 187-211, 10.2307/1942661, 1984.

Pennock, D. J.: Designing field studies in soil science, Canadian Journal of Soil Science, 84, 1-10, 10.4141/S03-039, 2004.

Schank, J. C., and Koehnle, T. J.: Pseudoreplication is a Pseudoproblem, Journal of Comparative Psychology, 123, 421-433, 10.1037/a0013579, 2009.

**1. Title page**

**No long-term effect of land-use activities on soil carbon dynamics in tropical montane**

**grasslands**

Viktoria Oliver[1,2*], Imma Oliveras[3], Jose Kala[4], Rebecca Lever[5,2], Yit Arn Teh [1,2]

[1] Institute of Biological and Environmental Sciences, University of Aberdeen, Cruickshank

Building, St. Machar Drive, AB24 3UU Aberdeen, UK.

[2] Formerly at the School of Geography and Geosciences, University of St Andrews, UK

[3] Environmental Change Institute, School of Geography and the Environment, University of

Oxford. South Parks Road, OX13QY Oxford, UK.

[4] Universidad de Santo Antonio Abad del Cusco, Cusco, Peru.

[5] Department of Life & Environmental Sciences, University of California, Merced 5200 North

Lake Rd. Merced, CA 95343, United States.

[*] Corresponding author: v.oliver@abdn.ac.uk

*Running title:* Tropical montane grassland soil carbon dynamics

*Keywords:* Andean montane grasslands, soil respiration, fire, grazing, puna, soil carbon, land- use activities, soil density fractionation.

**2. Abstract**

**2. Abstract**

Montane tropical soils are a large carbon (C) reservoir, acting as both a source and a sink of

$CO_2$. Enhanced $CO_2$ emissions originate, in large part, from the decomposition and losses of soil organic matter (SOM) following anthropogenic disturbances. Therefore, quantitative knowledge of the stabilization and decomposition of SOM is necessary in order to understand, assess and predict the impact of land management in the tropics. In particular, labile SOM is an early and sensitive indicator of how SOM responds to changes in land use and management practices, which could have major implications for long term carbon storage and rising atmospheric $CO_2$ concentrations. The aim of this study was to investigate the impacts of grazing and fire history on soil C dynamics in the Peruvian montane grasslands; an understudied ecosystem, which covers approximately a quarter of the land area in Peru. A

density fractionation method was used to quantify the labile and stable organic matter pools, along with soil $CO_2$ flux and decomposition measurements. Grazing and burning together significantly increased soil $CO_2$ fluxes and decomposition rates and reduced temperature as a driver. Although there was no significant effect of land use on total soil C stocks, the combination of burning and grazing decreased the proportion of C in the free LF, especially at the lower depths (10-20 and 20-30 cm). In the control soils, 20 % of the material recovered was in the free LF, which contained 30 % of the soil C content. In comparison, the burnt- grazed soil, had the smallest recovery of the free LF (10 %) and a significantly lower C content (14 %). The burnt soils had a much higher proportion of C in the occluded LF (12%) compared to the not-burnt soils (7%) and there was no significant difference among the treatments in the heavy F (~ 70%). The synergistic effect of burning and grazing caused changes to the soil

C dynamics. $CO_2$ fluxes were increased and the dominant temperature driver was obscured by some other process, such as changes in plant C and N allocation. In addition, the free LF

was negatively affected when these two anthropogenic activities took place on the same site.

Most likely a result of reduced detritus being incorporated into the soil. A positive finding from this study is that the total soil C stocks were not significantly affected and the long term (+10 years) C storage in the occluded LF and heavy F were not negatively impacted. Possibly this is because of low intensity fire, fire-resilient grasses and the grazing pressure is below the threshold to cause severe degradation.

**Commented [OV1]:** Reviewers comment: L.40-42. I would suggest including only percentage of soil C and not bulk soil to improve readability

% of bulk soil is correct but text changed to: "20 % of the material was recovered in the free LF"

**Commented [OV2]:** Reviewers comment: L.46-47: As autotropic respiration was not measured, I would omit these speculations in the abstract.

Autotrophic respiration removed from text

**Commented [OV3]:** Reviewers comment: L. 49-51: Please specify what you mean by long-term

~ 10+ years included in text

**3. Introduction**

High altitudinal montane grasslands (3200 - 4500 m a.s.l) account for a major proportion of land cover in the Andes, particularly in Peru, where they make-up approximately 25 % of land cover (Feeley and Silman 2010). Every year, especially in the dry season, large areas of these grasslands are burned to support traditional cattle grazing, which has been apparent since the early 1500s (Luteyn 1992). Fires for agricultural clearing and maintenance of these highly productive forage grasses is of considerable importance in these ecosystems and for the livelihood of the local people (Sarmiento and Frolich 2002). To some extent, this natural system is tolerant of these management practices (Ramsay 1992). However, in recent years, it has become apparent that the combination of global warming and the considerable pressure from agricultural expansion have resulted in increased fire occurrence and subsequent destruction of tropical montane cloud forest (Cochrane and Ryan 2009). Evidence of fire scars and charcoal deposits along the forest-puna tree line demonstrate a gradual encroachment into the adjacent tropical montane cloud forest (Di Pasquale *et al.* 2008).

Previous research in these Andean montane grasslands have measured large soil C stores, (Gibbon *et al.* 2010; Oliveras *et al.* 2014b). However, despite the concern on the effects of land management practices, there are very few studies on soil C dynamics in this tropical region of the Peruvian Andes. It is particularly unclear how land management affects the soil C dynamics and sequestration potential under the influence of grazing and burning. For example, (Oliveras *et al.* 2014b) , found that grazing and fire in montane grasslands resulted in decreased net primary productivity, but there were no differences between these two disturbances. Studies in other montane grasslands have found that an increase in the frequency of fire events can reduce the amount of soil organic matter (SOM) in the top soil (Knicker 2007), or it may increase the biomass growth period afterwards, causing more detritus to accumulate in the upper soil layers (Ojima *et al.* 1994).

SOM influences many soil functions and occupies a key position in the global C cycle (Lal 2004). It is a highly heterogeneous and dynamic composite of organic molecules (such as: polysaccharides, lignin, aliphatic biopolymers, tannins, lipids, proteins and aminosugars) derived from progressively decomposed plant, animal and microbial material (Zimmermann *et al.* 2007a; Totsche *et al.* 2010).

**Commented [OV4]:** Reviewers comment: L.58-65: How often are these grasslands burnt? Every 10-20 years, once for pasture establishment? How important is burning for these systems?

This paragraph now includes a bit more information about burning in puna systems

**Commented [OV5]:** Reviewers comment: L.69: What do you mean by soil C balances?

Changes from 'balance' to 'dynamics'

[revised manuscript text omitted]

**Commented [OV6]:** Reviewers comment: L.92-101: see general comment on new literature on SOC stability and ecosystem properties (e.g. Schmidt et al. 2011)
The cited literature could be improved: New literature and concepts about stability of SOC could improve the manuscript, such as Schmidt et al. 2011, Nature 478, 49-56. In addition, a literature overview about density fractions is missing. E.g. one tropical study is cited for many tropical, temperate and boreal studies. L.118. The same citation is used for a generally ranking of the results. L.-401-402. Including literature about density fraction and turnover times could improve the manuscript. On the other hand general statements (management history; L.60-61) are documented with 5 citations.

Literature review rewritten to include new literature on density fractions and turnover times

**Commented [OV7]:** Reviewers comment: L.124/L: Which particle-sizes were separated? Where are the results?

Particle size fractionation removed

**Commented [OV8]:** Reviewers comment: L.133: please specify different management systems
L.133-134: please specify labile and stable OM pools

Text changed to include management systems and OM pools

**Commented [OV9]:** Reviewers comment: L.135-137: Which environmental drivers do you mean except soil temperature and VWC? Please specify the objective

[revised manuscript text omitted]

**Commented [OV11]:** Reviewers comment: L.185: How were the bi-monthly measurements extrapolated to gain annual emissions?
What is the uncertainty of the annual emission? The annual emission is only based on 6 measurement days – without information on soil temperature course of the year. Soil respiration is driven by soil temperature (L270), but measurements only included day measurements at a very low frequency. What do you want to express with the annual emission rates?

Annual emissions removed

**Commented [OV12]:** Reviewers comment: L.203-208: Does the free LF included (living) roots or were they sorted out before?
(This would have major implications for the yield of free LF), see also comment L.299

Clarification that any living roots were removed from the soil was included in the text polycarbonate filter (Whatman Nuclepore Track Etch Membrane) to remove residual NaPT.

[revised manuscript text omitted]

**Commented [OV14]:** Reviewers comment: L.261-262: Does at Acjanaco grazing and burning significantly increase soil CO2 fluxes? From Figure 2, I do not get the impression.

Text changed to clarify that we mean grazing and burning significantly increase soil CO2 fluxes at wayqecha, not Acjanaco.

**Commented [OV15]:** Reviewers comment: L.269: How is season defined? By soil temperature and VWC? Are soil temperature and air temperature not strongly correlated?

Season was defined in the site descriptions section

**5.2 Decomposition rates**

[revised manuscript text omitted]

Alternatively, burning can cause significant losses of N due to combustion, and grasses may
compensate for increased N limitation by increasing their allocation to roots, thereby
increasing root respiration and potentially promoting enhanced belowground C cycling
(Johnson and Matchett 2001). Some evidence was found for this type of response in prior
work; Oliveras *et al.,* 2014, found higher below and above-ground C stocks in undisturbed
soils. While overall net primary productivity (NPP) was higher on undisturbed sites, NPP
belowground was greater with grazing and fire, suggesting a shift in plant allocation patterns
after these disturbances.

**Commented [OV17]:** Reviewers comment: L. 362. As heterotrophic respiration is not measured: may enhances.., as it is a speculation

'may enhance' included in text

**6.2 Belowground C stocks**

Overall, large total SOC stocks were measured in these montane grasslands (123 – 238 Mg C ha$^{-1}$), which is in keeping with other high elevation grassland studies and are probably attributable to low temperatures and wet conditions causing slow mineralisation of SOM and turnover rates. For example, in the Qinghai-Tibetan Plateau grasslands and páramo grasslands of the Colombian, Ecuadorian and Peruvian Andes, total SOC stocks can range between 80 – 250 Mg C ha$^{-1}$ (Hofstede 1995; Zimmermann *et al.* 2010; Farley *et al.* 2012; Li *et al.* 2013; Muñoz, Faz and Zornoza 2013; Oliveras *et al.* 2014b).

Soil C stocks were higher at Acjanaco than at Wayqecha. This is in agreement with Oliveras *et al.,* 2014, although the Acjanaco sites in this previous study were higher (253 compared to 175 Mg C ha$^{-1}$ reported here), perhaps reflecting within site spatial heterogeneity. There was no significant effect of either burning or grazing but grazing had a more negative effect than burning on the total soil C stocks. This negligible effect of burning may be a consequence of low intensity fires, fire-resilient grasses, and potentially low fuel loads at the time of burning (Knicker 2007). Grassland fires on slopes can move very quickly, so even when intense, the transfer of heat to the soil is less damaging due to low residence times (Rollins, Cohen and Durig 1993). As a result, surface temperatures do not typically exceed 100 °C or 50 °C at 5 cm depth (Campbell *et al.* 1995), and organic matter can only be fully volatilized between 200 and 315 °C (Knicker 2007). Even if the soils were dry at the time of burning which is possible during the dry season, then belowground temperatures would rise very slowly because of the insulating properties of air-filled pores, which curtail heat transfer belowground (Neary *et al.* 1999).

Grazing on the other hand, had a more negative impact on total SOC content than burning but there was not a significant loss of total soil C. One explanation is that the grazing pressure in these sites may have been below the threshold required to cause severe degradation, supporting previous studies in the Peruvian Andes, where they also found no significant effect of grazing or burning on total SOC stocks (Gibbon *et al.* 2010; Oliveras *et al.* 2014b).

**Commented [OV18]:** Reviewers comment: L.376: It would be nice to have a range of soil C stocks found in montane grassland soils

A range of C stocks from montane grasslands has been included in the text

**Commented [OV19]:** Reviewers comment: L.385-L399. There was no effect of burning on total soil C and no significant effect of grazing on total soil C. However grazing had a more negative effect on total soil C. please clarify.

Text changed to clarify that we mean both burning and grazing had no significant effect on total soil C

Overall, the free LF was larger than in other tropical systems (30 % of total soil C). By comparison, studies in Equador, Brazil and Puerto Rico found the free LF ranged from only 4-

12 % of total soil C content (Paul, Veldkamp and Flessa 2008; Marin-Spiotta *et al.* 2009; Potes

*et al.* 2012). However, it is difficult to compare the results of this study to other tropical fractionation studies because in general, most field sites are in tropical lowland pastures where soil C stocks tend to be lower. When comparing to other high elevational studies, for example, in permafrost meadow ecosystems in the Tibetan Qinghai Province, results are similar, with the free LF making up 27 % of the total soil C stocks (Dörfer *et al.* 2013).

Grazing had a negative impact on the free LF. As grazing is known for reducing aboveground biomass (Johnson and Matchett 2001; Gibbon *et al.* 2010), a lower incorporation of detritus into the soil is not surprising and has been observed in other grazing studies (Figueiredo,

Resck and Carneiro 2010; Cao *et al.* 2013). The effects of grazing on the free LF were most pronounced when grazing and burning occurred together, in which case, the free LF showed the most pronounced declines. To our knowledge there are also no other studies assessing the impact of grazing and burning on soil C fractions in high altitude tropical grasslands.

When measuring the soil organic pools, the long-term effects of land-use can be gained by relatively short-term experiments because burning, in theory, could have a relatively immediate impact on all the pools of carbon. In this study, the significant positive effect of burning on the occluded LF may be the results of charcoal particles (from burning) becoming incorporated into the occluded LF. Charcoal, because of its low density, tends to reside in the lighter fractions (Cadisch *et al.* 1996; Glaser *et al.* 2000; Sollins *et al.* 2006), despite its recalcitrance. Because the fires took place almost ten years ago, the charcoal may no longer be resident the free LF but may have become occluded into soil micro-aggregates due to its high sorptive capacity (Qayyum *et al.* 2014). Once incorporated into micro-aggregates, charcoal can be maintained for centuries after fire (Zackrisson, Nilsson and Wardle 1996).

**7. Conclusions**

This study highlights the complexities of how land management can affect soil C dynamics in montane tropical grasslands. The results suggest that montane grasslands are resilient to soil
* * *
**Commented [OV20]:** Reviewers comment: L.401-411: Please expand literature. In addition, please check the number cited (10%) and carefully consider the land use type.
It would be nice to have the range of free LF found in tropical soils in order torank and interpret the gained results (L 403-406).

% and the type of land use is correct.
Additional free LF data from other studies has been included for comparison

**Commented [OV21]:** Reviewers comment: L.413-420: Does this mean that burning favours long-term stabilisation of soil C as charcoal? It is it is stated (L49-51), that the long term storage in the occluded fraction was not negatively impacted, but has a positive effect?

Question answered in authors response and text changed to include a sentence on long-term effects in short-term experiments

[revised manuscript text omitted]

**Commented [OV23]:** Reviewers comment: Table 1: I would like to have the given information (BD, pH C:N, Soil C ) at least for both sites and different depth (and management system). For me it is not clear which soil is descripted in Table 1.

More information on soil properties included for each plot

**Table 2** Annual and seasonal mean soil temperature, VWC and $CO_2$ flux for Wayqecha and Acjanaco for each land management system. Different letters down the columns represent significant differences between sites.

| Site / land use | Soil temp. (°C) at 5 cm | VWC (%) at 5 cm | $CO_2$ flux ($\mu$mol m$^{-2}$ s$^{-1}$) |
|---|---|---|---|
| **Wayqecha (2003)** | 14.7 ± 0.1 | 62.3 ± 0.4 | 1.31 ± 0.09 |
| Grazed – burnt | 15.3 ± 0.3[a] | 63.4 ± 0.3[ab] | 1.88 ± 0.23[a] |
| Grazed - not burnt | 14.5 ± 0.2[ab] | 63.8 ± 0.2[ab] | 1.07 ± 0.07[b] |
| Not grazed - burnt | 14.6 ± 0.3[ab] | 60.9 ± 1.0[c] | 0.99 ± 0.08[bc] |
| Not grazed - not burnt | 14.1 ± 0.2[b] | 62.5 ± 0.8[bc] | 1.10 ± 0.07[ab] |
| Dry season | 14.1 ± 0.2 | 61.4 ± 0.8 | 1.35 ± 0.16 |
| Wet season | 15.1 ± 0.20 | 63.8 ± 0.3 | 1.31 ± 0.10 |
| Minimum | 11.6 | 29.9 | 0.22 |
| Maximum | 18 | 65.8 | 8.33 |
| **Acjanaco (2005)** | 11.6 ± 0.1 | 64. 5 ± 0.1 | 0.91 ± 0.03 |
| Grazed – burnt | 12.0 ± 0.2[c] | 64.0 ± 0.2[ab] | 0.82 ± 0.05[bc] |
| Grazed - not burnt | 11.5 ± 0.2[cd] | 64.5 ± 0.2[ab] | 0.84 ± 0.07[bc] |
| Not grazed - burnt | 11.9 ± 0.1[cd] | 64.2 ± 0.2[ab] | 0.77 ± 0.05[c] |
| Not grazed - not burnt | 10.8 ± 0.1[d] | 65.1 ± 0.2[a] | 0.72 ± 0.05[c] |
| Dry season | 11.6 ± 0.1 | 63.8 ± 0.2 | 0.81 ± 0.04 |
| Wet season | 11.7 ± 0.1 | 65.1 ± 0.1 | 0.74 ± 0.03 |
| Minimum | 9.5 | 57.1 | 0.09 |
| Maximum | 13.7 | 67.7 | 2.69 |
| GRAZED – BURNT | 13.8 ± 0.2[a] | 63.7 ± 0.2[a] | 1.35 ± 0.13[a] |
| GRAZED – NOT BURNT | 13.2 ± 0.2[a] | 64.1 ± 0.1[a] | 0.95 ± 0.05[b] |
| NOT GRAZED – BURNT | 13.3 ± 0.2[a] | 62.6 ± 0.5[a] | 0.88 ± 0.05[b] |
| NOT GRAZED – NOT BURNT | 12.6 ± 0.2[a] | 63.8 ± 0.4[a] | 0.91 ± 0.05[b] |

**Table 3**. Mean soil C content (Mg C ha[-1]) for each depth and total C stocks (0-30 and 0-20 cm) on all the land management systems. Different letters down the columns within each depth represent significant differences among sites. All values are given with 1 standard error of the mean (n = 3).

[Figure]

[Figure]

**Figure 4** Mass losses (%) of sticks from the decomposition experiment (*n* = 3) on two burnt sites (2003 = Wayqecha and 2005 = Acjanaco) with grazed subplots and control plots.

**Table 4** Mean mass recovery of density fractions and proportion of total C residing in the three density fractions (%) from the total soil profile (0-30 cm). Different letters down the columns represent significant differences.

| | Free LF | | Occluded LF | | Heavy F | |
|---|---|---|---|---|---|---|
| | Fraction of total C (%) | Mass of soil recovered (%) | Fraction of total C (%) | Mass of soil recovered (%) | Fraction of total C (%) | Mass of soil recovered (%) |
| Grazed - burnt | 14.0 ± 5.3[b] | 9.9 ± 3.6[a] | 10.8 ± 2.6[ab] | 9.8 ± 3.4[ab] | 76.0 ± 8.0[a] | 78.4 ± 7.2[a] |
| Not grazed - burnt | 19.7 ± 8.3[ab] | 15.1 ± 8.5[a] | 14.2 ± 2.5[a] | 11.3 ± 4.7[a] | 66.1 ± 10.5[a] | 76.6 ± 8.3[a] |
| Grazed - not burnt | 22.7 ± 13.3[ab] | 16.2 ± 8.5[a] | 8.9 ± 2.1[bc] | 5.3 ± 1.6[bc] | 68.3 ± 14.0[a] | 76.7 ± 8.1[a] |
| Not grazed - not burnt | 30.0 ± 5.7[a] | 19.5 ± 5.5[a] | 5.2 ± 0.8[c] | 4.3 ± 0.7[c] | 64.7 ± 6.1[a] | 69.7 ± 5.8[a] |

[Figure]

**Figure 5** Mass of soil recovered in the three density fractions (%) on the four left bar plots and the proportion of total C residing in the three density fractions (%) on the four right bar plots for the different land uses (a = 0-5 cm, b = 5-10 cm, c = 10-20 cm, d = 20-30 cm). Error bars indicate 1 standard error of the mean ($n$ = 6).

---

## Author Response (AR2)

**DETAILED RESPONSE TO REFEREES**

On behalf of my co-authors, I would like to thank the editor for his constructive comments on our manuscript. A detailed description of how we have responded to your comments is provided below.

**RESPONSE TO EDITOR**

1. Both referees suggest changing the manuscript title, which was not done. The title is too unspecific concerning the land use activities. In addition in my view the study cannot evaluate the long-term effect of burning and grazing on soil dynamics. I agree with the authors stating that burning can have immediately effects on stable pools. However, that the stable pool is not immediately affected does not mean that there are no long-term effects: There were changes in the active and in the resistant pool (change in free LF and occluded LF). How these changes will affect the inert pool on long term is not clear. (In this meaning there were no short term effects on inert pool ........). Please clarify the title. Please check also L446-448.

**Author's response:** The manuscript title has been changed to: "The effects of burning and grazing on soil carbon dynamics in managed Peruvian tropical montane grasslands".

2. In addition, regarding the uppers 0-5 cm (Figure 5) there seem to be differences in the fraction of total soil C in the heavy fraction: The control site free LF as the most dominant soil C fraction, while in the grazed burnt plot most carbon is stored in the heavy fraction. The "inert" pool is altered. One explanation may be that the concept of stability of pools does not fit or that there are also effects of burning / grazing on the inert pool. However, differences in the soil C fraction in the first 5cm are not discussed at all. Although it could be assumed the changes due to land use are most pronounced in the upper part of the soil (highest temperature while burning, highest input of SOM). The same trend, less pronounced could be observed in 5-10 cm. Please make a concise interpretation of all the data included in the manuscript.

**Author's response:** The authors agree with your comment and the results section has now be changed to include a description of the surfaces soil layers for all the C pools. L351-370

3. The authors provided more information about pool concept of different stability in soils. However, the wording is a bit confusing. The three different pools are introduced as active, recalcitrant, inert (L102). The recalcitrant pool is than named resistant pool (L109) or as "slow recalcitrant". Please use stringent definitions.

**Author's response:** The three different pools have now been changed to be consistently the same throughout the manuscript (active, resistant and inert). L103

4. I do not like the judgemental wording of negative or positive effects / finding throughout the manuscript, e.g. L 46 "the free light fraction was negatively affected". "the free light fraction was reduced" is much more specific without any subjective judgement. E.g. L 47-49; L347, L428. In addition the occluded LF increased, but it is stated that this fraction was not negatively impacted. According to the concept that would meant that the fraction of soil C in a resistant pool increased and thus in the authors sense would be positively affected.

**Author's response:** The negative and positive wording throughout the manuscript have been amended to be more specific and without subjective judgement (L333-334, L351, L358).

5. L123-124. This sentence is not nicely embedded into paragraph.

**Author's response:** This sentence has now been incorporated earlier in the paragraph (L119-120).

6. L293: In the results section the unit of soil respiration is adapted to the measuring frame (express per seconds), while during discussion section the unit is still per year (L357) suggesting an annual database ( or even a yearly emission). I would prefer to use the measuring frame of seconds throughout the manuscript.

**Author's response:** The manuscript has now been changed to include the measuring frame in per seconds in the discussion but in parentheses the annual units have been kept, with a disclaimer to inform the reader that that this is purely for a general comparison and that we do not have sufficient data to provide accurate yearly emission estimates (L347-381).

6. L 434: In the discussion of proportion of LF to total soil C, only one reference is given. For a comparison of the dataset, this is insufficient – especially there are reviews available (like Gregorich, E.G., Beare, M.H., Mckim, U.F. & Skjemstad, J.O. 2006. Chemical and biological characteristics of physically uncomplexed organic matter. Soil Science Society of America Journal, 70, 975–985) and more recent original papers. Please expand literature review.

**Author's response:**

The suggested review paper has been included and additional recent literature has been added into the discussion (L461-464, L468-475).

7. Table 4 and Figure 5: It would be nice to have the same order of fraction of C and Mass recovery

**Author's response:** The order of fraction of C and Mass recovery has been changed.

9. Table 3 is missing

**Author's response:** Table 3 has now been included.

10. Figure 3: heading is missing. It would be nice to add the names of the sites to 2003 and 2005.

**Author's response:** Heading and names of the sites have been added.

11. L656: The reference is not complete.

**Author's response:** Reference has been completed